# Similarity Is Not Logic: Factored Inference for Dual-Encoder Vision-Language Models

**Sultan Alshehri**[1]  **Zhantao Yang**[1]  **Han Zhang**[1]  **Marios Savvides**[1]

## Abstract

Dual-encoder vision–language models (VLMs) expose a similarity interface that enables zero-shot retrieval but fails compositional constraints: queries like "umbrella and no person" retrieve images containing both, even when concept detection is reliable. We trace this to an interface-level **Bag-of-Concepts** effect, where similarity scores approximate mean pooling of concept evidence regardless of operators. Although operator-dependent signals exist in text embeddings, they are too weak or misaligned to affect rankings. Fine-tuning does not reliably resolve this failure because the dominant bottleneck is how similarity aggregates evidence rather than what encoders represent. We propose **factored inference**, which separates evidence extraction from constraint execution, and introduce LCSE (Logic-Constrained Score Editing), a training-free method that executes constraints externally using concept scores from frozen encoders. We also introduce FACTOR-Bench, where LCSE achieves 85.5% accuracy versus 73.2% for the best fine-tuned baseline, 90.7% when applied to SigLIP 2, and improves NegBench COCO MCQ accuracy from 27.2% to 65.2% while preserving retrieval performance.

## 1. Introduction

Contrastive dual-encoder vision–language models (VLMs) such as CLIP (Radford et al., 2021) and SigLIP (Zhai et al., 2023) map images and text into a shared embedding space and expose a single matching primitive: a scalar similarity. This similarity is highly effective for zero-shot retrieval and is increasingly used as a *control surface* for structured querying in real pipelines. However, a scalar similarity

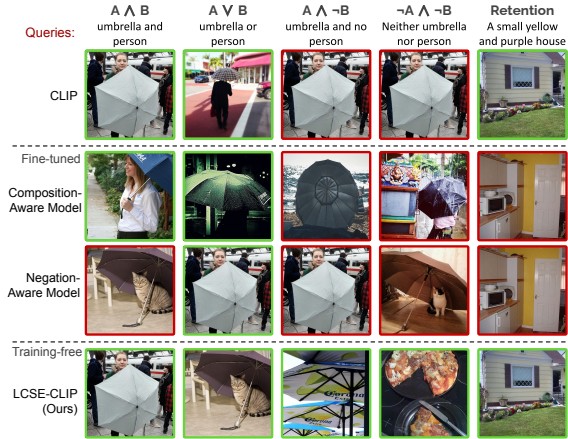

*Figure 1.* **CLIP-style retrieval models violate compositional constraints.** Negation and exclusion queries retrieve excluded concepts, and conjunctions miss required concepts. CLIP and its composition-aware and negation-aware (Yuksekgonul et al., 2023; Alhamoud et al., 2025) variants show unreliable top-1 retrieval results. LCSE (§4) reduces these failures while preserving standard retrieval performance.

is not a *constraint language*. When queries specify how evidence should be combined—e.g., negation, conjunction, or exclusion—the resulting rankings can violate intended truth-functional semantics.

Figure 1 illustrates this failure. Queries such as "*umbrella and no person*" or "*neither umbrella nor person*" cause CLIP-style models to retrieve images containing the excluded concepts. Prior approaches (Alhamoud et al., 2025; Park et al., 2025; Singh et al., 2025) fine-tune on negation-augmented captions and hard negatives, but struggle to preserve standard retrieval behavior and alter rankings outside the negation/operator settings. Across COCO concept pairs (Lin et al., 2014) and in practical retrieval systems, exclusion constraints systematically retrieve what they aim to exclude (§3). This reflects a core limitation: dual-encoder similarity behaves like a bag-of-concepts model, approximately mean-pooling evidence for individual concepts rather than executing compositional operators (§3).

Boolean operators provide a clean setting because satisfaction criteria are unambiguous (e.g., $A \wedge B$ is satisfied iff both concepts are present), which enables direct evaluation

[1]Carnegie Mellon University, Pittsburgh, PA, USA. Correspondence to: Sultan Alshehri <salshehr@andrew.cmu.edu>.

of whether an interface executes intended semantics rather than simply correlating with them (Alhamoud et al., 2025; Zhang et al., 2025; Zhou et al., 2025). This motivates us to (i) isolate the mechanism behind these failures in current dual-encoder VLMs, and (ii) design an interface-level fix that requires no retraining and preserves standard retrieval performance.

Failures may arise from (i) *representation* (operator-dependent information is absent from text embeddings), or (ii) *execution* (such information exists but does not affect rankings under dot-product scoring). We disentangle these factors in §3. Across models, operator-dependent directions are detectable in text embeddings (Quantmeyer et al., 2024), but are attenuated or misaligned under dot-product scoring. As a result, compound prompts behave like soft pooling of atomic concept evidence and track *which* concepts are mentioned more than *how* they are combined.

**We propose factored inference and logic-constrained score editing (LCSE).** Our *factored inference* interface separates *evidence extraction* from *constraint execution*, given a parsed query (concepts, polarities, operator). We extract an atomic evidence score per concept using the frozen VLM and execute the Boolean operators externally. To preserve non-constraint aspects captured by standard similarity (e.g., scene context and style), we apply a lightweight *score edit* where LCSE adaptively modifies the similarity score only when the operators require non-mean aggregation (§4). This paper makes the following contributions:

1. **Mechanistic analysis of constraint failure.** We provide controlled diagnostics showing that Boolean operators are not executed reliably under scalar similarity, and we isolate the mechanism where dot-product scoring is dominated by concept evidence and behaves like soft pooling, while operator-dependent signals are too weak or misaligned to enforce truth-functional constraints (§3).

2. **A training-free interface for constraint execution.** We propose **LCSE**, a factored inference interface that executes Boolean constraints explicitly over atomic evidence while preserving standard retrieval behavior where no constraints apply (§4).

3. **FACTOR-Bench.**[1] We introduce **FACTOR-Bench**, a benchmark for Boolean operator semantics built from operator-contrastive caption pairs with validated concept presence/absence. It enables controlled measurement of whether a model distinguishes operator semantics or merely tracks concept mentions (§5).

4. **Experimental validation across models and benchmarks.** We evaluate LCSE on FACTOR-Bench, Neg-

---

[1]Project page, code, and benchmark: https://sultanmo.github.io/factored-vlm/.

Bench (Alhamoud et al., 2025), and retrieval retention tasks against standard CLIP and fine-tuned variants, where we observe consistent improvements across constraint types and backbones (§5). We empirically validate our claims on Boolean operators over one–two visual concepts.

## 2. Related Work

**Dual-encoder VLMs and similarity-based retrieval.** Contrastive dual-encoder vision–language models such as CLIP (Radford et al., 2021), ALIGN (Jia et al., 2021), LiT (Zhai et al., 2022), SigLIP (Zhai et al., 2023), and SigLIP 2 (Tschannen et al., 2025) learn aligned image and text embeddings that enable zero-shot retrieval and classification with a fixed matching rule (cosine similarity / inner product). This *similarity interface* is widely reused as a plug-in primitive in downstream systems and evaluations (Schrodi et al., 2025).

**Compositionality benchmarks and bag-of-words behavior.** A substantial line of work shows that strong average retrieval performance does not imply robust compositional understanding. Benchmarks such as ARO (Yuksekgonul et al., 2023), Winoground (Thrush et al., 2022), CREPE (Ma et al., 2023), and VL-CheckList (Zhao et al., 2022) probe sensitivity to structure, attribute binding, and relations, and often find bag-of-words-like failure modes. Later work highlights that compositional evaluations can be hackable (Hsieh et al., 2023; Kamath et al., 2024; Udandarao et al., 2025), and training-based mitigations fine-tune with structured hard negatives (Yuksekgonul et al., 2023; Peleg et al., 2025). Probing suggests compositional structure exists in encoders but is not reliably reflected in cross-modal similarity (Lewis et al., 2024; Koishigarina et al., 2025).

**Negation and logical operators in VLMs.** Neg-Bench (Alhamoud et al., 2025) provides systematic negation diagnostics for retrieval and multiple-choice settings, and NegVQA (Zhang et al., 2025) studies negation in VQA-style formulations. More broadly, recent benchmarks and analyses investigate logical failures of vision–language systems beyond negation alone (Zhou et al., 2025). Mitigation efforts are largely training-based and use hard/counterfactual negatives, paired affirmative–negated captions, or logic-aware objectives (e.g., data-centric negation fine-tuning in Neg-Bench (Alhamoud et al., 2025), CoN-CLIP (Singh et al., 2025), NegationCLIP (Park et al., 2025), and logic-aware training such as LogicCLIP (Zhou et al., 2025)). Other works, such as (Quantmeyer et al., 2024), probe CLIP through causal tracing to find how negation is represented.

**Interface-level matching and compositional scoring.** Given evidence that compositional signals exist but are suppressed under global similarity, a complementary di-

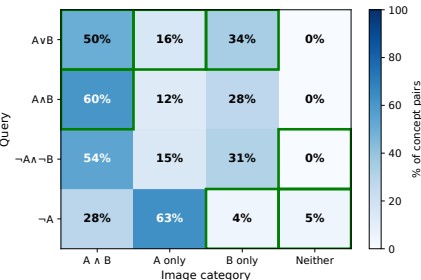

*Figure 2.* **Holistic scoring inverts anti-monotone operators.** Cells show % of concept pairs where each image category achieves the highest mean similarity score. Green borders indicate expected (correct) categories. $\neg A \wedge \neg B$ should prefer NEITHER but *never* does (0%). $\neg A$ prefers images containing $A$ for 91% of pairs. $A \wedge B$: BOTH wins only 60%.

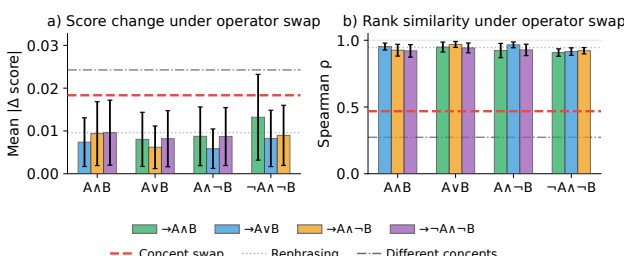

*Figure 3.* **Content words dominate while operators have weak effect.** Swapping operators ("and" ↔ "or") largely preserves rankings ($\rho \approx 0.93$), while swapping concepts substantially reorders them ($\rho \approx 0.64$), consistent with a bag-of-concepts interface.

rection modifies the matching *interface* rather than only the encoders. DCSM (Kang et al., 2025) constructs dense patch–token similarity maps and learns a lightweight scoring network over these maps to enable more expressive matching than a single global dot product. ComCLIP (Jiang et al., 2024) disentangles images into sub-components and composes matching scores without retraining.

**Modular and neuro-symbolic visual reasoning.** A long-standing theme in multimodal reasoning is to separate perception from structured computation, including neural module networks and program-based approaches. Recent LLM-driven visual programming systems execute explicit intermediate steps to improve compositional reliability (Gupta & Kembhavi, 2023; Surís et al., 2023), and logical reasoning pipelines often separate perception, premise selection, and rule execution (Xu et al., 2025). These systems typically introduce new perceptual modules, learned executors, or multi-step reasoning loops.

Prior work addresses compositional failures through encoder fine-tuning or external reasoning. Fine-tuning targets the wrong bottleneck, while external reasoning diverges from the similarity interface. Our analysis shows the encoder often extracts reliable atomic evidence. In this regime the failure is primarily in how dot-product scoring combines it (§3), which we quantify by stratifying gains by evidence quality (§5.3). We fix this interface-level bottleneck directly by extracting evidence with the frozen encoder and executing constraints externally while preserving both correct semantics and retrieval quality.

## 3. Motivation

Dual-encoder VLMs compress an image-text match into a single scalar similarity. This is sufficient for many zero-shot tasks, but logical operators impose *compositional constraints* on rankings: truth-functional requirements that holistic scoring does not satisfy reliably. For example, a

negated query should lower the score of images containing the negated concept, and a conjunction should rank images missing any conjunct below those satisfying all. Logical constraints provide a controlled case study with unambiguous satisfaction criteria (e.g., conjunction requires all conjuncts present), which enables precise measurement of whether the interface executes the intended semantics.

**Setup.** Throughout this section, we use CLIP ViT-B/32 (Radford et al., 2021) evaluated on COCO val2017 (Lin et al., 2014). We refer to standard dual-encoder similarity $s(I, T) = \langle f(I), g(T) \rangle$ as *holistic scoring*. For two-concept queries, we partition images into four quadrants based on ground-truth annotations: BOTH (both concepts present), A-ONLY, B-ONLY, and NEITHER. We proceed in three steps: first, document the empirical failure pattern, then isolate a mechanism via span-residual decomposition, and finally examine why training-based fixes do not address it.

**Holistic scoring ignores logical structure.** We compare rankings for *"a X"* versus *"no X"* across object categories. CLIP reliably detects whether $X$ is present (AUC = 0.88, i.e., $P(s_{X\text{-present}} > s_{X\text{-absent}})$). Yet the two queries produce nearly identical rankings (Spearman $\rho = 0.94$), which indicates that negation has almost no effect on image ordering. Although negation does change the text embedding, it does not flip rankings: a failure of *execution* rather than absence of representation.

For two-concept queries, we ask which quadrant attains the highest mean holistic score. Figure 2 summarizes this *quadrant preference*. Anti-monotone queries invert systematically: $\neg A \wedge \neg B$ (NOR) assigns highest scores to images containing *both* concepts (should prefer neither). Similarly, $\neg A$ assigns highest scores to images where the negated concept is *present* (should prefer absent). Conjunction is also unstable. For $A \wedge B$, BOTH wins only 60% of pairs, while A-ONLY/B-ONLY win the remainder. In contrast, $A \vee B$

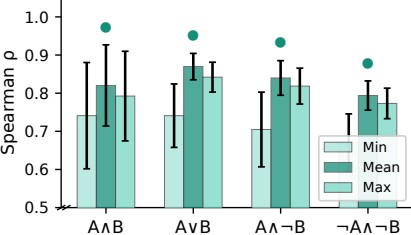

*Figure 4.* **Holistic scores approximate mean pooling.** Across operators, scores correlate most strongly with $\text{MEAN}(s_A, s_B)$, not truth-functional aggregators. Even $\neg A \land \neg B$ (neither), which should anti-correlate with both concepts, is predicted by their mean.

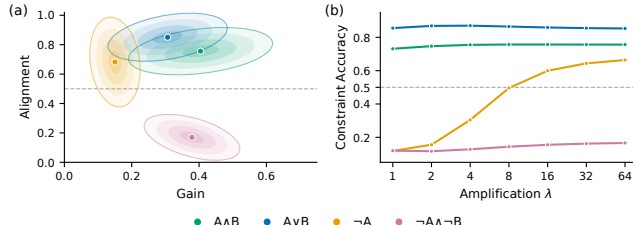

*Figure 5.* **Two failure regimes.** (a) Gain vs. alignment ($2\sigma$ ellipses over concept pairs, dashed line = chance). $A \land B$, $A \lor B$, and $\neg A$ cluster above chance (aligned), while $\neg A \land \neg B$ clusters far below (misaligned). (b) Amplification sweep ($\lambda$): as residual influence increases, scores converge toward their alignment ceiling. $\neg A$ improves dramatically ($12\% \rightarrow 66\%$), while $\neg A \land \neg B$ remains inverted.

appears less problematic because evidence accumulation is monotone and often compatible with inclusive disjunction. Figure 3 tests whether operators affect image rankings. Trivial rephrasing ($\rho = 0.95$) and unrelated queries ($\rho = 0.27$) establish upper and lower bounds on ranking similarity. Operator swaps fall near the upper bound ($\rho = 0.94$), so changing "and" to "or" has no more effect than rewording. Only concept replacement ("cat and dog" $\rightarrow$ "bird and dog") causes rankings to diverge ($\rho = 0.47$). The model attends to concept content while ignoring logical structure. This bag-of-concepts behavior strengthens the bag-of-words framing (Yuksekgonul et al., 2023), which treats text as an unordered set of contributing tokens. Here, operator tokens effectively drop out, and compound scores depend only on which concepts are present.

**Mechanism: span-residual decomposition.** The preceding failures share a pattern. Operators that should *decrease* scores instead *increase* them. We separate two questions: (i) *representation*, whether operator-dependent information exists in the text embedding, and (ii) *execution*, whether that information influences image rankings under dot-product scoring. We show that compound embeddings are dominated by concept information, with operator-specific signal present but weak or misaligned.

Given a compound query $T$ combining concepts $A$ and $B$, let $\mathbf{t} = g(T)$ be the text embedding, $\mathbf{v} = f(I)$ the image embedding, and $\mathbf{t}_A, \mathbf{t}_B$ the embeddings of atomic prompts "a $A$" and "a $B$" (so $s_A = \langle \mathbf{v}, \mathbf{t}_A \rangle$, $s_B = \langle \mathbf{v}, \mathbf{t}_B \rangle$). We decompose $\mathbf{t}$ into a **span component** (linear in atomic embeddings) and an **orthogonal residual** (operator-specific):

$$\mathbf{t} = \mathbf{t}_{\text{span}} + \mathbf{t}_{\perp}, \qquad (1)$$

where $\mathbf{t}_{\text{span}} = \text{proj}_{\text{span}(\mathbf{t}_A, \mathbf{t}_B)}(\mathbf{t})$ and $\mathbf{t}_{\perp} = \mathbf{t} - \mathbf{t}_{\text{span}}$ is the orthogonal residual. Under dot-product scoring, this induces a score decomposition: $\langle \mathbf{v}, \mathbf{t} \rangle = \langle \mathbf{v}, \mathbf{t}_{\text{span}} \rangle + \langle \mathbf{v}, \mathbf{t}_{\perp} \rangle$, a span term (linear in atomic scores $s_A, s_B$) plus a residual term (operator-specific). Because the span can only produce linear combinations of atomic scores, it behaves like pooling, and any nonlinear operator semantics (min, max,

complement) must come from the residual.

**The span dominates and approximates mean pooling.** Figure 4 shows that holistic scores for all four operators are well-approximated by $\text{MEAN}(s_A, s_B)$ rather than truth-functional aggregators: $A \land B$ correlates more strongly with mean ($\rho = 0.82$) than min ($\rho = 0.74$). Even $\neg A \land \neg B$ is predicted by the mean of concepts it should exclude. Direct computation of the span coefficients ($\mathbf{t}_{\text{span}} = \alpha \mathbf{t}_A + \omega \mathbf{t}_B$) yields approximately balanced weights: $\alpha/(\alpha + \omega) \approx 0.51$–$0.53$ across two-concept operators, consistent with mean rather than a skewed linear combination (Appendix D).

**Operator gain and alignment.** To quantify the residual's contribution, let $z_{\text{span}}(I) = \mathbf{t}_{\text{span}}^{\top} \mathbf{v}$ and $z_{\perp}(I) = \mathbf{t}_{\perp}^{\top} \mathbf{v}$ be the span and residual score components for image $I$. We define *operator gain* as the variance fraction attributable to the residual:

$$G = \frac{\text{Var}_I(z_{\perp}(I))}{\text{Var}_I(z_{\text{span}}(I)) + \text{Var}_I(z_{\perp}(I))}. \qquad (2)$$

Low $G$ implies the residual rarely flips rankings relative to span-only scoring (Proposition 1 Appendix E discusses the required independence assumption. *Operator alignment* measures whether the residual points in the correct direction: the AUC when ranking by $z_{\perp}$ alone. For $A \land B$, alignment is $P(z_{\perp}(I_{\text{BOTH}}) > z_{\perp}(I_{\text{other}}))$—the probability that a BOTH image outscores a violating image (50% is random, 100% is perfect). We measure constraint accuracy as the AUC at each amplification level $\lambda$, where at $\lambda{=}1$ this equals holistic AUC, and as $\lambda \rightarrow \infty$ it converges to alignment.

Figure 5(a) maps these metrics across concept pairs. The key distinction is *alignment*: operators above the dashed line (50%) have residuals encoding correct semantics, while those below are inverted. Gain determines how much the residual influences rankings at $\lambda{=}1$. Low gain means the span dominates and requires amplification to recover the residual's signal. Figure 5(b) tests whether amplifying the residual via $\mathbf{t}_{\lambda} = \mathbf{t}_{\text{span}} + \lambda \mathbf{t}_{\perp}$ can recover operator signals.

Since $\mathbf{t} = \mathbf{t}_{\text{span}} + \mathbf{t}_\perp$ by construction, $\lambda{=}1$ recovers holistic similarity exactly, and as $\lambda$ increases, the residual dominates and scores converge toward alignment.

**Two failure regimes** emerge from the figure.

*Regime A (aligned):* $A \wedge B$, $A \vee B$, and $\neg A$. Residuals encode correct operator semantics (alignment 68–85%). For conjunction and disjunction, holistic scoring already achieves 73–86% AUC because the span (mean-pooling) correlates with satisfaction, and amplification yields marginal improvement. For negation, holistic scoring achieves only 12% AUC, the span dominates and acts like positive evidence for the negated concept. However, the residual is correctly aligned (68%), so amplification recovers performance toward $\approx 66\%$.

*Regime B (misaligned):* $\neg A \wedge \neg B$. The residual has consistent structure but points in the wrong direction, and alignment is 17%, far below random. Amplification cannot help because the residual encodes "concepts are salient", since it correlates *positively* with concept presence rather than absence. Null baselines confirm that residuals encode structured signal rather than noise (Appendix C). Shuffling residuals across concepts collapses alignment to chance, and the negation direction $\Delta_X = g(\text{``not } X\text{''}) - g(\text{``}X\text{''})$ is consistent across concepts (cosine 0.62) with correct polarity (AUC = 0.70), which confirms that negation is encoded but not executed.

**Proposition 1** (Rank stability). *A rank inversion between span-only and full scoring on pair $(I, J)$ requires $|\Delta z_{span}| < |\Delta z_\perp|$ with opposite signs. Under Gaussian pairwise differences, the expected inversion rate is bounded by a monotone function of $\sigma_\perp / \sigma_{span}$ and vanishes as $G \to 0$.*

When gain is low, the residual does not overcome the span, so rankings follow mean-like aggregation regardless of operator. Across all four operators, residuals encode structured, operator-specific information, but dot-product scoring fails to leverage it. For aligned operators ($A \wedge B$, $A \vee B$, $\neg A$), the span already dominates rankings, while for NOR, the residual itself encodes concept salience rather than joint exclusion.

**Why training fixes are brittle under a scalar interface.** The interface-level bottleneck predicts that fine-tuning the encoder should not fix Boolean failures, since the problem is how scores aggregate evidence, not what the encoder represents. We test this on several negation-focused CLIP variants. Table 1 applies a polarity test to multiple negation-focused variants on COCO val2017 (Lin et al., 2014). For query "no $X$," correct negation requires images *without $X$* to outscore images *with $X$* (AUC > 50%). All six fine-tuned models achieve below-chance AUC. Negation still behaves like positive evidence. Fine-tuning may improve some benchmark metrics, but it does not induce truth-functional

*Table 1.* **Polarity test.** AUC for "no $X$": P(image without $X$ outscores image with $X$). All tested holistic models (including fine-tuned variants) score below chance. Factored scoring applies negation externally and achieves correct polarity.

| Model | AUC ↑ |
|---|---|
| CLIP (Radford et al., 2021) | 11.2% |
| NegCLIP (Yuksekgonul et al., 2023) | 9.3% |
| CoN-CLIP (Singh et al., 2025) | 17.2% |
| CLIP-NegFull (Alhamoud et al., 2025) | 12.7% |
| NegCLIP-NegFull (Alhamoud et al., 2025) | 10.9% |
| NegationCLIP (Park et al., 2025) | 19.0% |
| Factored $(1{-}p)$ | **88.7%** |

negation in the similarity interface. In contrast, a simple *factored* approach (extract concept evidence $p$ from the frozen encoder, apply negation externally as $1{-}p$) achieves correct polarity by construction (Table 1, last row).

Since recovering operator semantics from holistic similarity has limited success even with fine-tuning, we next develop a factored approach that separates evidence extraction from constraint execution.

## 4. Method

The analysis in §3 shows that holistic similarity conflates concept evidence with constraint enforcement, which produces approximately $\text{MEAN}(s_A, s_B)$ regardless of the operator connecting $A$ and $B$. This explains why negation fails to reverse rankings (the score has no reliable way to "subtract" evidence) and why conjunction enforcement is unreliable (a conjunctive constraint depends on the weakest required evidence, not an average).

We therefore aim for an interface that has: *compositional correctness* (C1), which requires truth-functional constraints rather than implicit averaging, and *no degradation* (C2), which requires that retrieval quality be preserved where holistic similarity already works. The core principle of our approach is to separate what the VLM does well (evidence extraction) from what holistic scoring fails to execute (constraint composition).

Given a query $A \wedge \neg B$, we (1) parse it into concepts ($A$: present, $B$: absent) under conjunction, (2) score each concept independently using the frozen encoder, (3) apply polarity ($1{-}p$ for absent) and aggregate via power mean, and (4) blend with the holistic score to preserve non-constraint aspects (scene, style).

We achieve C1 and C2 without retraining (Algorithm 1):

$$p(I, T) = \underbrace{\mathcal{C}}_{\text{constraint}} \Big( \underbrace{\mathcal{E}(I, T)}_{\text{evidence}} \Big), \tag{3}$$

where $\mathcal{E}$ extracts atomic evidence scores from the image–text pair, and $\mathcal{C}$ executes the compositional constraint over these scores. This template is not inherently limited to logical constraints. Different compositional queries instantiate different evidence extractors and constraint executors. For logical constraints over visual concepts, we instantiate:

- $\mathcal{E}$: concept presence scores via the frozen encoder (§4.2)
- $\mathcal{C}$: power-mean aggregators ($\approx \min / \max$) plus complement $1-(\cdot)$

We additionally introduce *score editing* (§4.3) to preserve holistic information for non-constraint aspects (C2).

## 4.1. Problem Setup

Let $f$ and $g$ be the frozen image and text encoders with $\ell_2$-normalized outputs. The *holistic similarity* $s_{\text{hol}}(I,T) \triangleq \langle f(I), g(T) \rangle$ is calibrated to $(0,1)$ via sigmoid: $p_{\text{hol}}(I,T) \triangleq \text{sigm}\big(\beta(s_{\text{hol}}(I,T) - \mu)\big)$, where $(\mu, \beta)$ are constants estimated from held-out data (Appendix A).

Given a caption $T$, we parse it into a structured representation $\pi(T) = \big(\{(c_i, n_i)\}_{i=1}^m, o\big)$, where each $c_i$ is a visual concept phrase, $n_i \in \{0, 1\}$ indicates negation ($n_i{=}1$ means *absent*), and $o$ is the operator (conjunction, disjunction, or none). Complex operators use polarity flags, e.g., $A \wedge \neg B$ parses as conjunction over $A$ (present) and $B$ (absent). We use an LLM with structured outputs to extract this parse. All parses are pre-computed and cached for reproducibility (Appendix A). This parse makes explicit what holistic similarity conflates: the *concepts* being queried and the *operator* combining them.

## 4.2. Factored Scoring

We instantiate $\mathcal{E}$ and $\mathcal{C}$ for logical constraints over visual concepts. The VLM provides concept detection and constraints are executed externally.

For each concept $c_i$, we compute a calibrated presence score:
$$p_i \triangleq \text{sigm}\big(\beta(s_i - \mu)\big) \in (0,1), \qquad (4)$$
where $s_i = \langle f(I), g(c_i) \rangle$ is the image-concept similarity and $(\mu, \beta)$ are the calibration constants from §4.1.

**Polarity.** §3 showed that negation does not flip rankings in holistic similarity ($\rho = 0.94$ between "a $X$" and "no $X$"). We apply negation as a complement:
$$\tilde{p}_i = \begin{cases} 1 - p_i & \text{if } n_i = 1 \text{ (absent)} \\ p_i & \text{if } n_i = 0 \text{ (present)} \end{cases} \qquad (5)$$

This implements truth-functional negation. If $p_i$ is the evidence that $c_i$ is present, then $1-p_i$ is the evidence it is absent. The complement ensures perfect rank reversal ($\rho = -1$) under strictly monotone calibration, up to ties. Compound

patterns like $A \wedge \neg B$ and $\neg A \wedge \neg B$ (NOR) are handled by applying per-concept polarity before aggregation.

**Constraint execution.** Given polarity-adjusted evidence $\tilde{\mathbf{p}} = (\tilde{p}_1, \ldots, \tilde{p}_m)$, the constraint executor $\mathcal{C}$ aggregates according to the operator $o$. Truth-functional semantics require conjunction to be bounded by the minimum (all conjuncts must hold) and disjunction by the maximum (any disjunct suffices). We implement these via the power mean, $M_\gamma(\mathbf{p}) = \big(\frac{1}{m} \sum_i p_i^\gamma\big)^{1/\gamma}$, which interpolates smoothly between $\min$ ($\gamma \to -\infty$), arithmetic mean ($\gamma = 1$), and $\max$ ($\gamma \to +\infty$):
$$p_{\text{logic}} \triangleq M_{\gamma_o}(\tilde{\mathbf{p}}), \qquad (6)$$
where $\gamma_o < 0$ for conjunction, $\gamma_o \gg 1$ for disjunction, and $\gamma_o = 1$ otherwise. This soft relaxation improves robustness to calibration noise while preserving the correct ordering constraints (Appendix G reports values and alternatives). Factored scoring thus replaces the implicit mean-pooling of holistic similarity with constraint-appropriate aggregation.

## 4.3. Score Editing (LCSE)

Factored scoring satisfies C1 but risks degrading retrieval. A simpler alternative—using $p_{\text{logic}}$ when operators are detected and falling back to $s_{\text{hol}}$ otherwise—discards holistic information entirely for operator queries. However, $p_{\text{hol}}$ encodes more than concept presence (scene context, spatial layout, stylistic cues), and discarding it hurts performance on natural-language queries where this context matters (Appendix H). LCSE instead preserves holistic information while adjusting only for operator-specific aggregation.

$$s_{\text{LCSE}} = s_{\text{hol}} + \frac{1}{\beta} \underbrace{\big(\text{logit}(p_{\text{logic}}) - \text{logit}(p_{\text{soft}})\big)}_{\text{correction}}, \qquad (7)$$

where $\text{logit}(p) = \log \frac{p}{1-p}$ maps probabilities back to similarity space via the inverse calibration, and:

- $p_{\text{soft}} = \frac{1}{m} \sum_{i=1}^m p_i$ is the mean of calibrated concept scores (reference for simple averaging)

- $p_{\text{logic}} = M_{\gamma_o}(\tilde{\mathbf{p}})$ aggregates *polarity-adjusted* scores via power mean (Eq. 6)

The correction measures how much constraint-appropriate aggregation differs from simple averaging, projected back to similarity space via the inverse calibration. This adjusts $s_{\text{hol}}$, pushing it down when constraints require stricter conditions (e.g., conjunction with a weak conjunct), up when they are more permissive (e.g., disjunction with a strong disjunct), and leaving it unchanged when no adjustment is needed. When no logical operators apply, $p_{\text{logic}} = p_{\text{soft}}$ (both reduce to the mean), so the correction vanishes and $s_{\text{LCSE}} = s_{\text{hol}}$. Algorithm 1 gives the full procedure.

**Algorithm 1** Factored inference with score editing (LCSE).

---

**Require:** image $I$, caption $T$, frozen encoders $f, g$
 1: **Parse:** $\{(c_i, n_i)\}_{i=1}^{m}, o \leftarrow \pi(T)$
 2: **Holistic:** $s_{\text{hol}} \leftarrow \langle f(I), g(T) \rangle$
 3: **Evidence $\mathcal{E}$:**
 4: **for** $i = 1$ to $m$ **do**
 5: $\quad s_i \leftarrow \langle f(I), g(c_i) \rangle$
 6: $\quad p_i \leftarrow \text{sig}(\beta(s_i - \mu))$
 7: $\quad \tilde{p}_i \leftarrow (1 - n_i)\, p_i + n_i\, (1 - p_i)$ $\qquad \triangleright$ Eq. 5
 8: **end for**
 9: **Constraint $\mathcal{C}$:** $p_{\text{logic}} \leftarrow M_{\gamma_o}(\tilde{\mathbf{p}})$ $\qquad \triangleright$ Eq. 6
10: **Score edit:** $p_{\text{soft}} \leftarrow \frac{1}{m} \sum_i p_i$
11: **return** $s_{\text{hol}} + \frac{1}{\beta}\big(\text{logit}(p_{\text{logic}}) - \text{logit}(p_{\text{soft}})\big)$

---

## 5. Experiments

We evaluate LCSE on three complementary axes: (1) boolean operator accuracy on FACTOR-Bench, (2) negation handling on NegBench (Alhamoud et al., 2025), and (3) retention of zero-shot retrieval quality. All methods use ViT-B/32 except DCSM (Kang et al., 2025) (ViT-B/16).

### 5.1. Experimental Setup

**Baselines.** We compare against standard holistic similarity interfaces (**CLIP** (Radford et al., 2021), **SigLIP 2** (Tschannen et al., 2025)) and several fine-tuned alternatives: **Neg-CLIP** (Yuksekgonul et al., 2023) (fine-tuned with hard negatives), **CoN-CLIP** (Singh et al., 2025) (fine-tuned on negation-augmented captions), **NegationCLIP** (Park et al., 2025) (finetuned on LLM-generated negation pairs), **DCSM** (Kang et al., 2025) (replaced cosine similarity with a learned CNN over patch-token similarity maps), and **CLIP-NegFull/NegCLIP-NegFull** (CC12M (Changpinyo et al., 2021) models fine-tuned with NegBench negation data).

**Benchmarks. FACTOR-Bench** (ours) tests whether scoring interfaces execute Boolean operator semantics correctly across five constraint types: negation ($\neg A$), conjunction ($A \wedge B$), disjunction ($A \vee B$), exclusion ($A \wedge \neg B$), and NOR ($\neg A \wedge \neg B$). Each sample uses a *pairwise* format. Given an image, choose between two captions where one satisfies the constraint and one violates it. The key design is *operator-contrastive pairs*: captions sharing the same concepts but differing in logical operators. We show a sample:

> **Image:** car present, bicycle absent
> $\wedge$ Conjunction: "car" ✓ | "car and bicycle" ✗
> $\neg$ Negation: "no bicycle" ✓ | "bicycle" ✗
> $\vee$ Disjunction: "car or bicycle" ✓ | "car and bicycle" ✗

This design directly tests operator understanding: if a model assigns similar scores to "car or bicycle" and "car and bicycle," it reveals bag-of-concepts behavior. The model tracks *which* concepts appear rather than *how* they combine. To ensure valid measurement, we use balanced polarity ($\sim$50%

negation correct) to prevent text-only shortcuts, template diversity (5+ phrasings per operator) to prevent lexical memorization, and OWL-ViT (Minderer et al., 2022) validation of concept presence/absence for ground-truth accuracy. The main benchmark is a 1,100-sample *operator* split covering the five constraint types above (NOT, AND, OR, BUT-NOT, NEITHER). FACTOR-Bench includes two supplementary splits, a 450-sample *equivalence* split (De Morgan, double negation, commutativity) and a 145-sample *compound* split (3–4 concepts, mixed polarity), for 1,695 samples in total. Oracle parses are embedded for reproducibility (Appendix B). **NegBench** (Alhamoud et al., 2025) provides 4-way MCQ and retrieval on COCO (Lin et al., 2014), and 4-way MCQ on VOC2007 (Everingham et al., 2010) (for cross-benchmark evaluation), for comparison with prior work on negation understanding. **Retention** measures zero-shot retrieval preservation (R@5 and Spearman $\rho$) with CLIP and is evaluated on COCO and PASCAL Sentences Dataset (Rashtchian et al., 2010), where each of 5 captions per image serves as an independent query over all 1000 images.

Our method, **LCSE** (Logic-Constrained Score Editing) extracts atomic concept evidence and applies truth-functional corrections only where logical constraints require it (§4). Given a caption, an LLM parser (GPT-4.1-mini (OpenAI et al., 2024)) first extracts and caches concept phrases and polarity markers. Each concept is scored using frozen encoders via template ensembling (averaging over prompts like "a $c$", "a photo of a $c$"), then calibrated with backbone-specific parameters $(\mu, \beta) = (0.22, 30)$ for CLIP-family backbones and $(0.05, 30)$ for SigLIP 2. Calibration parameters are estimated on COCO train2017 (as few as $\sim$50 held-out images suffice for CLIP, see Appendix A) and reused across evaluation benchmarks without recalibrating. These scores are then aggregated via power mean: $\gamma = -1$ for conjunction, $\gamma = 10$ for disjunction, $\gamma = 1$ otherwise. On COCO val captions, 97.5% parse without explicit operators or negation, where the LCSE correction vanishes and scoring reduces to holistic.

### 5.2. Main Results

Table 2 presents our main findings. On CLIP ViT-B/32, LCSE achieves 85.5% FACTOR-Bench accuracy vs. 73.2% for the best baseline (NegationCLIP), and 86.2% when stacked with NegCLIP. Unlike fine-tuned models, which show substantial rank displacement ($\rho = 0.63$–$0.85$), LCSE maintains standard retrieval within 0.1 percentage points of CLIP (55.2% vs. 55.3% R@5 on COCO, $\rho = 0.999$, 82.4% R@5 on PASCAL, $\rho = 0.999$). On NegBench MCQ, LCSE improves over NegCLIP-NegFull on both COCO (60.3% vs. 56.2%) and VOC2007 (73.3% vs. 59.6%).

The method generalizes across backbones. Applied to

*Table 2.* **Constraint satisfaction and zero-shot retention.** FACTOR-Bench tests Boolean operators; NegBench tests negation MCQ (%) and retrieval R@5 (%); Retention measures R@5 (%) and $\rho$ with CLIP. **Bold**: best among CLIP methods; baseline ; ours . [†] denotes models using a different backbone.

| | FACTOR | NegBench | | | Retention | | | |
| | | COCO | | VOC2007 | COCO | | PASCAL | |
| Method | Acc↑ (50%) | MCQ↑ (25%) | R@5↑ | MCQ↑ (25%) | R@5↑ | $\rho$ | R@5↑ | $\rho$ |
|---|---|---|---|---|---|---|---|---|
| CLIP (Radford et al., 2021) | 58.3 | 39.3 | 48.0 | 38.7 | 55.3 | 1.00 | 82.4 | 1.00 |
| SigLIP 2 [†] (Tschannen et al., 2025) | 65.0 | 27.2 | 64.1 | 27.1 | 73.5 | — | 93.2 | — |
| DCSM (Kang et al., 2025) | 53.6 | 44.8 | 21.8 | 44.2 | 34.6 | .63 | 64.2 | .65 |
| CoN-CLIP (Singh et al., 2025) | 63.2 | 25.7 | 48.3 | 39.7 | 52.3 | .79 | 79.4 | .79 |
| NegCLIP (Yuksekgonul et al., 2023) | 67.2 | 28.7 | 64.4 | 30.5 | 69.3 | .76 | 89.8 | .77 |
| CLIP-NegFull (Alhamoud et al., 2025) | 66.5 | 54.5 | 51.9 | 54.8 | 54.7 | .84 | 81.3 | .85 |
| NegCLIP-NegFull (Alhamoud et al., 2025) | 68.1 | 56.2 | **67.1** | 59.6 | **69.6** | .80 | **90.0** | .81 |
| NegationCLIP (Park et al., 2025) | 73.2 | 31.0 | 65.8 | 42.1 | 68.6 | .81 | 89.8 | .83 |
| LCSE (CLIP) | 85.5 | **60.3** | 50.8 | 73.3 | 55.2 | **.999** | 82.4 | **.999** |
| LCSE (SigLIP 2) [†] | 90.7 | 65.2 | 67.1 | 87.2 | 73.5 | — | 93.1 | — |
| LCSE (NegCLIP) | **86.2** | 55.7 | 65.0 | **81.3** | 69.3 | .76 | 89.8 | .77 |

*Table 3.* **Per-constraint accuracy on FACTOR-Bench (%).** Anti-monotone constraints (negation, NOR) require polarity inversion, which holistic scoring does not provide.

| | Anti-monotone | | | Monotone | |
| | $\neg A$ | $\neg A \wedge \neg B$ | $A \wedge \neg B$ | $A \wedge B$ | $A \vee B$ |
|---|---|---|---|---|---|
| CLIP | 58.0 | 54.4 | 66.0 | 68.7 | 42.0 |
| SigLIP 2 [†] | 59.7 | 55.2 | 68.8 | 76.7 | 74.0 |
| DCSM | 55.7 | 58.4 | 53.6 | 50.0 | 45.3 |
| CoN-CLIP | 67.7 | 52.4 | 61.2 | 86.7 | 52.0 |
| NegCLIP | 66.0 | 55.2 | 71.6 | 96.0 | 53.3 |
| CLIP-NegFull | 73.7 | 78.4 | 73.2 | 68.7 | 19.3 |
| NegCLIP-NegFull | 77.3 | 70.8 | 74.0 | 76.0 | 27.3 |
| NegationCLIP | 76.3 | 58.8 | 79.2 | 91.3 | 62.7 |
| LCSE (CLIP) | **88.0** | **82.4** | **82.8** | 84.7 | 90.7 |
| LCSE (SigLIP 2) [†] | 92.7 | 88.8 | 86.4 | 89.3 | 98.7 |
| LCSE (NegCLIP) | **88.0** | 76.8 | 81.2 | **96.7** | **96.0** |

*Table 4.* **LCSE gain scales with concept-detection quality.** FACTOR-Bench operator split stratified by the minimum per-concept detection AUC among a sample's concepts. LCSE exceeds holistic in every bin; the gain is largest where atomic evidence is reliable.

| Min concept AUC | $n$ | **Holistic** | **LCSE** | $\Delta$ |
|---|---|---|---|---|
| High ($> 0.90$) | 417 | 57.8 | **90.4** | +32.6 |
| Medium (0.75–0.90) | 479 | 61.4 | 85.2 | +23.8 |
| Low ($< 0.75$) | 204 | 52.0 | 76.0 | +24.0 |
| Overall | 1100 | 58.3 | 85.5 | +27.2 |

SigLIP 2, LCSE reaches 90.7% on FACTOR-Bench and 65.2% on NegBench MCQ, demonstrating that better atomic evidence directly translates to better constraint execution.

Table 3 breaks down accuracy by constraint type. CLIP achieves only 54–66% on anti-monotone constraints, while LCSE (CLIP) reaches 82–88% across the same operators. LCSE disjunction accuracy is high (91–99%) because its constraint is compatible with max aggregation, which evidence accumulation approximates. CLIP's low disjunction accuracy (42%) reflects FACTOR-Bench's operator-contrastive pairs ("A or B" vs "A and B"), where holistic scoring assigns near-identical scores.

### 5.3. When Does LCSE Help? Stratifying by Evidence Quality

Our mechanistic analysis attributes constraint failures to the scoring interface rather than the encoder, and this holds

where the encoder supplies reliable atomic evidence. We probe this boundary by stratifying the FACTOR-Bench operator split by the weakest per-concept detection AUC in each sample (the binding factor, since the correction aggregates all concept scores). Table 4 shows that LCSE's gain scales with evidence quality. With strong detection (AUC $> 0.90$) LCSE reaches 90.4% (vs. 57.8% holistic). Under weak detection (AUC $< 0.75$), it degrades to 76.0% (still +24pp over holistic). This supports an *interface-dominant* reading of §3, where if reliable evidence is available, external execution nearly closes the gap, and residual errors increasingly reflect evidence extraction rather than operator execution.

### 5.4. Ablation Studies

We ablate calibration, aggregation, and the LCSE correction term in Appendices A, G, and H. Table 5 shows LCSE improves anti-monotone constraints for every backbone tested, with gains from +3.3 to +28.1 points. The improvement is largest for SigLIP 2 (+28.1) and CLIP (+24.9), both lacking negation-aware training. Fine-tuned variants gain less (NegationCLIP +3.3, NegCLIP +17.7), so LCSE complements backbone improvements.

*Table 5.* **LCSE improves all backbones on anti-monotone constraints.** Average accuracy (%) on anti-monotone operators ($\neg A$, $\neg A \wedge \neg B$, $A \wedge \neg B$).

| Backbone | Holistic | +LCSE |
|---|---|---|
| SigLIP 2 | 61.2 | **89.3** (+28.1) |
| CLIP | 59.5 | 84.4 (+24.9) |
| NegCLIP | 64.3 | 82.0 (+17.7) |
| CLIP-NegFull | 75.1 | 87.2 (+12.1) |
| NegCLIP-NegFull | 74.0 | 80.9 (+6.9) |
| CoN-CLIP | 60.4 | 66.7 (+6.3) |
| NegationCLIP | 71.4 | 74.7 (+3.3) |

## 6. Conclusion

Compositional queries require factored inference: separating evidence extraction from constraint execution. We demonstrated this principle on logical constraints over visual concepts, where standard holistic scoring exhibits a *Bag-of-Concepts* problem. Scores track which concepts are mentioned, not how they are combined. The model *can* detect atomic concepts, but it does not combine them correctly under the standard scoring interface. Our method, LCSE, addresses this by extracting atomic evidence separately, executing operators externally, and enforcing truth-functional semantics, substantially improving task performance (85.5% FACTOR-Bench accuracy vs. 73.2% best baseline, 60.3% NegBench MCQ vs. 56.2%) while maintaining standard retrieval (55.2% vs. 55.3% R@5, $\rho = 0.999$), all without retraining the encoders. The method generalizes across architectures: applied to SigLIP 2, LCSE reaches 90.7% on FACTOR-Bench, which shows that better atomic evidence directly translates to better constraint execution.

Our analysis and method are empirically validated on Boolean operators over visual concepts, a controlled setting with unambiguous ground truth. The failure mechanism (soft evidence aggregation under scalar similarity) and the fix (external constraint execution) are demonstrated in this setting. Within this scope, factored inference also handles *nested* Boolean structure via recursive aggregation over the parse tree (e.g., 3-concept queries: holistic 51.0% vs. LCSE 80.5%; Appendix I). Whether this principle extends to other compositional structures (e.g., attribute binding, relations, counting) depends on the availability of reliable atomic evidence, a question we leave to future work. As VLMs are increasingly used for structured querying and filtering, understanding interface-level limitations becomes critical.

**Limitations.** Our method depends on parsing Boolean structure, though substituting real GPT-4.1-mini parses for oracle parses changes FACTOR-Bench accuracy by only $-0.3$pp, and a missed operator makes the correction vanish so that scoring defaults to holistic (Appendix A). The approach requires $m+1$ text encodings per query (where $m$ is the number of concepts), though pre-caching concept embeddings reduces retrieval-time overhead to $\sim 4\%$ wall-clock under a fused implementation, stable up to 500K-image pools (Appendix K). If the encoder fails to produce atomic evidence for a concept, external operator execution does not recover correctness. LCSE shifts failures from operator execution to evidence extraction, as the evidence-quality stratification confirms (§5.3).

## Impact Statement

This paper presents work whose goal is to advance the field of machine learning. There are many potential societal consequences of our work, none of which we feel must be specifically highlighted here.

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

# A. Implementation Details

**Caption parser.** We parse captions using GPT-4.1-mini with structured JSON outputs. The output schema enforces:

- `concepts`: list of `{text, is_negated}` pairs
- `operator`: one of SINGLE (single concept, $m=1$), AND (explicit conjunction), OR (explicit disjunction), or NONE (multiple concepts without explicit connector, $m>1$)

The parser uses 83 few-shot examples and is prompted with detailed rules for: (i) distinguishing explicit conjunction ("a dog and a cat" → AND) from unconstrained queries ("a dog chasing a cat" → NONE), (ii) detecting negation patterns including explicit ("no X", "without X"), implicit ("X-free", "lacking X"), and compound ("neither X nor Y" → both negated with AND), (iii) handling false positives such as quantifiers ("no fewer than 3 dogs" → affirmed) and compound nouns ("no smoking sign" → affirmed), (iv) merging subject phrases with verbs and attributes ("tall man running" as one concept).

Table 7 shows representative parses. All parses are pre-computed and cached, and evaluation uses only cached parses for reproducibility.

**Benchmark-specific parsing.** FACTOR-Bench includes oracle parses (ground truth) embedded in the benchmark, which isolates LCSE evaluation from parser errors. NegBench and COCO retention tests use LLM parser outputs. On FACTOR-Bench, we evaluate the LLM parser against oracle parses across 3,171 captions: concept extraction achieves 99.9% accuracy (3,169/3,171), and operator classification achieves 97.5% accuracy (3,091/3,171). Operator errors are largely due to "alongside" phrases classified as NONE rather than AND. The parser achieves near-complete coverage: FACTOR-Bench 100% (3,171/3,171), NegBench 99.8% (34,210/34,264), and COCO 99.7% (24,733/24,808). Missing captions are malformed entries (empty strings, meta-comments from data generation).

**Parser robustness on natural language.** The parser accuracy reported above is measured against FACTOR-Bench oracle parses, which use minimal templates. For natural language captions (e.g., COCO val), direct accuracy measurement is less meaningful because ground-truth parses are not uniquely defined—"a woman carrying bananas" could reasonably parse as one or two concepts. However, three factors mitigate parser sensitivity on natural language: (1) FACTOR-Bench evaluation uses oracle parses, so all FACTOR-Bench accuracy results are parser-independent. (2) NegBench captions are LLM-generated with predictable patterns ("A $X$ is present, but no $Y$") that align with the parser's 83 few-shot examples. (3) On COCO val, only 2.5%

*Table 6.* **End-to-end LCSE accuracy: oracle vs. LLM parses** (FACTOR-Bench operator split, CLIP ViT-B/32). Substituting real GPT-4.1-mini parses for oracle parses costs only $-0.3$pp overall; no parse error induces a wrong-sign correction.

|  | $\neg A$ | $A \wedge B$ | $A \vee B$ | $A \wedge \neg B$ | $\neg A \wedge \neg B$ | **All** |
|---|---|---|---|---|---|---|
| Holistic CLIP | 58.0 | 68.7 | 42.0 | 66.0 | 54.4 | 58.3 |
| LCSE (oracle) | 88.0 | 84.7 | 90.7 | 82.8 | 82.4 | **85.5** |
| LCSE (LLM) | 88.0 | 82.7 | 90.7 | 82.8 | 82.4 | **85.2** |

| Caption | Concepts | Op |
|---|---|---|
| `a dog and a cat` | dog, cat | AND |
| `a dog chasing a cat` | dog chasing, cat | NONE |
| `no dog` | dog$^{\neg}$ | SINGLE |
| `dog but not cat` | dog, cat$^{\neg}$ | AND |
| `neither dog nor cat` | dog$^{\neg}$, cat$^{\neg}$ | AND |
| `a man with a beard` | man with beard | SINGLE |
| `no smoking sign` | no smoking sign | SINGLE |

$^{\neg}$ = is_negated: true

*Table 7.* Representative caption parses. The parser distinguishes explicit logical connectors (AND/OR) from unconstrained queries (NONE), and correctly handles negation patterns, and avoids false positives on compound nouns.

of captions contain explicit Boolean operators (AND/OR) or negation, and for the remaining 97.5%, the LCSE correction vanishes and scoring reduces to holistic.

**End-to-end accuracy under real parses.** The accuracy figures in the main paper use FACTOR-Bench oracle parses to isolate execution from parsing. To measure the end-to-end cost of real parsing, we re-evaluate the operator split with GPT-4.1-mini parses (Table 6). Overall LCSE accuracy changes by only $-0.3$pp (oracle 85.5% → LLM 85.2%), with four of five operators unchanged; the gap comes from a handful of misparsed captions, and in every case the correction vanishes and scoring defaults to holistic (no wrong-sign correction is applied). This end-to-end accuracy is distinct from the parser's *operator-classification* accuracy (97.5%, above) and from its stricter *full-parse* match (concepts, polarity, and operator jointly correct), which is 91% over the operator split.

**Calibration parameters.** The sigmoid calibration $p = \sigma(\beta(s - \mu))$ converts raw cosine similarities $s \in [-1, 1]$ to probabilities $p \in (0, 1)$. The two parameters serve distinct roles in LCSE:

**Center** $\mu$ approximates the decision boundary between present and absent concepts: scores above $\mu$ map to $p > 0.5$ (present), scores below map to $p < 0.5$ (absent). This directly affects negation: the LCSE correction for $\neg A$ is $\frac{1}{\beta}\left(\text{logit}(1-p_A) - \text{logit}(p_A)\right) = -\frac{2}{\beta}\text{logit}(p_A)$, which is positive (boosting the score) when $p_A < 0.5$ and negative otherwise. Thus, $\mu$ determines the *sign* of the correction—

wrong $\mu$ causes negation to fail. Empirically, $\mu$ aligns with the median similarity of present concepts: $\mu=0.22$ for CLIP and $\mu=0.05$ for SigLIP 2.

**Slope** $\beta$ controls the spread of probabilities: small $\beta$ compresses all values toward 0.5, while large $\beta$ pushes them toward 0 or 1. This affects the *magnitude* of corrections. For conjunction $A \wedge B$, the correction is $\frac{1}{\beta}\big(\text{logit}(\text{HM}(p_A, p_B)) - \text{logit}(\text{AM}(p_A, p_B))\big)$, where HM is harmonic mean and AM is arithmetic mean. When $\beta$ is small, $p_A \approx p_B \approx 0.5$, so HM $\approx$ AM and the correction vanishes—LCSE reduces to holistic scoring. Larger $\beta$ spreads the probabilities, and enables meaningful corrections.

**Empirical calibration.** Both $\mu$ and $\beta$ are calibrated empirically on COCO train2017, which is held out from our val2017-based evaluations. We construct pairwise operator tests analogous to FACTOR-Bench. For each operator (NOT, AND, OR, BUT-NOT, NEITHER), we sample image-caption pairs where ground truth is determined by COCO object annotations. Grid search over $(\mu, \beta)$ maximizes average LCSE accuracy across all operators. This yields $(\mu=0.22, \beta=30)$ for CLIP and $(\mu=0.05, \beta=30)$ for SigLIP 2. Performance is stable across $\beta \in [30, 60]$ with $\leq 1\%$ variation, so we use $\beta=30$ as a conservative default. Recalibration is inexpensive. The grid search over $(\mu, \beta)$ on a small held-out set completes in seconds on a single GPU, as few as $\sim 50$ labeled images suffice to recover $\mu=0.22$ for CLIP, and the recovered $(\mu, \beta)$ transfer across all evaluation benchmarks without per-dataset retuning.

Tables 8 and 9 show ablation results on FACTOR-Bench (val2017). For both models, $\mu$ primarily affects negation (NOT varies 58–93%), while $\beta$ affects conjunction and disjunction (AND varies 75–90%, OR varies 83–99%). NOT and BUT-NOT are stable across $\beta$. For NOT, only the sign of the correction matters, and for BUT-NOT, the polarity flip $(1-p_B)$ already creates sufficient contrast between present and absent concepts, independent of $\beta$. In contrast, AND and OR require $\beta$ to spread probabilities apart so that harmonic/soft-max aggregation differs meaningfully from the arithmetic mean.

## B. FACTOR-Bench Design

FACTOR-Bench is a diagnostic benchmark for evaluating Boolean operator semantics in vision-language scoring interfaces. This section details its design principles, construction methodology, and anti-shortcut mechanisms.

The benchmark tests a specific question: *given correctly identified concepts, can the scoring interface execute Boolean constraints?* Unlike benchmarks that embed negation in complex natural-language sentences, FACTOR-Bench uses minimal templates to isolate operator semantics

*Table 8.* CLIP calibration ablation on FACTOR-Bench (%). $\mu$ affects negation, $\beta$ affects conjunction/disjunction. Bold: defaults ($\mu=0.22$, $\beta=30$).

| Param | Value | $\neg A$ | $A \wedge B$ | $A \vee B$ | $A \wedge \neg B$ | All |
|---|---|---|---|---|---|---|
| | 0.18 | 58.3 | 76.0 | 87.3 | 66.0 | 68.5 |
| | 0.20 | 77.7 | 81.3 | 88.7 | 82.0 | 81.1 |
| $\mu$ | **0.22** | **88.0** | 84.7 | 90.7 | 82.8 | **85.5** |
| | 0.24 | 87.7 | 86.0 | 92.7 | 76.8 | 83.4 |
| | 0.26 | 81.3 | 86.0 | 94.0 | 71.2 | 78.1 |
| | 10 | 88.0 | 75.3 | 83.3 | 82.8 | 82.8 |
| $\beta$ | **30** | **88.0** | 84.7 | 90.7 | 82.8 | **85.5** |
| | 50 | 88.0 | 86.0 | 94.0 | 82.0 | 86.2 |
| | 60 | 88.0 | 85.3 | 94.0 | 82.0 | 85.9 |

*Table 9.* SigLIP 2 calibration ablation on FACTOR-Bench (%). Bold: defaults ($\mu=0.05$, $\beta=30$).

| Param | Value | $\neg A$ | $A \wedge B$ | $A \vee B$ | $A \wedge \neg B$ | All |
|---|---|---|---|---|---|---|
| | 0.03 | 85.0 | 86.7 | 98.7 | 84.4 | 87.3 |
| | 0.04 | 90.3 | 88.7 | 98.7 | 86.8 | 90.6 |
| $\mu$ | **0.05** | **92.7** | 89.3 | 98.7 | 86.8 | **90.7** |
| | 0.06 | 92.3 | 89.3 | 98.7 | 83.2 | 88.9 |
| | 0.07 | 89.7 | 89.3 | 98.7 | 78.0 | 85.7 |
| | 10 | 92.7 | 81.3 | 98.0 | 86.0 | 88.5 |
| $\beta$ | **30** | **92.7** | 89.3 | 98.7 | 86.8 | **90.7** |
| | 50 | 92.7 | 90.0 | 98.7 | 86.4 | 91.0 |
| | 60 | 92.7 | 90.0 | 98.0 | 86.8 | 91.1 |

from language complexity. The pairwise format (choose between two captions for one image) directly measures constraint satisfaction without the confounds of top-$k$ ranking. The benchmark covers five Boolean constraint types, that span both monotone and anti-monotone operators:

| Operator | Constraint | Samples | Test condition |
|---|---|---|---|
| Negation | $\neg A$ | 300 | A absent |
| Conjunction | $A \wedge B$ | 150 | one present, one absent |
| Disjunction | $A \vee B$ | 150 | one present, one absent |
| Exclusion | $A \wedge \neg B$ | 250 | A present, B absent |
| NOR | $\neg A \wedge \neg B$ | 250 | both absent |
| Equivalence tests | | 450 | (see below) |
| Compound (3–4 concept) | | 145 | mixed polarity |

**Construction methodology.** Samples are constructed from COCO val2017 using the following procedure:

1. **Concept selection**: We use 80 COCO object categories, excluding "person" (too ubiquitous). Concept pairs are sampled with constraints: max 5 samples per pair, max 10 samples per concept.
2. **Image selection**: For each sample, we select images satisfying the required presence/absence conditions using COCO annotations.
3. **OWL-ViT validation**: We validate ground truth using OWL-ViT (Minderer et al., 2022) object detection. Images where the detector contradicts COCO annotations are rejected. This filters annotation noise and ensures

reliable ground truth.

4. **Caption generation**: Captions use minimal templates (e.g., "a {A} and a {B}", "no {A}") with 5+ phrasings per operator to prevent lexical memorization.

5. **Oracle parse embedding**: Each sample includes pre-computed ground-truth parses for evaluation independent of parser accuracy.

**Anti-shortcut mechanisms.** Three mechanisms prevent trivial solutions:

- **Balanced polarity**: For negation tests, ~50% have the negated caption correct and ~50% have the affirmed caption correct. A text-only baseline that always prefers affirmation (or always prefers negation) achieves only 50%.
- **Template diversity**: Each operator uses 5+ syntactic variants (e.g., "no X", "without X", "X-less scene", "lacking X"). Models cannot rely on specific lexical patterns.
- **Position randomization**: The correct answer appears in position A or B with roughly equal frequency.

**Equivalence tests.** FACTOR-Bench includes 450 samples testing logical equivalences as shortcut detectors:

- **De Morgan** (50 samples): "neither A nor B" ≡ "not A and not B"
- **Double negation** (100 samples): "X is not missing" ≡ "X is present"
- **Commutativity** (300 samples): "A and B" ≡ "B and A"

For these samples, both captions are logically equivalent (correct answer is "both"). Models using shortcuts (e.g., position bias, lexical heuristics) will assign different scores to equivalent formulas and expose inconsistent behavior. Section J analyzes violation rates across models.

## C. Null Baselines: Validating the Span-Residual Decomposition

The span-residual decomposition (§3) claims that the residual $\mathbf{t}_\perp$ encodes operator-specific information. We validate this claim through three lines of investigation: (1) whether the residual carries real signal rather than noise, (2) whether negation is encoded in the text representation, and (3) what mechanism drives the systematic inversion observed for NOR.

### C.1. Is the Residual Signal Real?

We first establish that residuals encode structured, concept-specific information rather than random noise.

**Comparison to random directions.** For each concept $X$, we generate 1000 random unit vectors orthogonal to $\mathbf{t}_X$ and compute their alignment (AUC for constraint satisfaction). We compare the actual residual's alignment to this null distribution. For negation ($\neg A$, 77 concepts): actual alignment $0.65 \pm 0.15$, null mean 0.50, percentile 80%, $z = 1.2$. For NOR ($\neg A \wedge \neg B$, 200 pairs): actual alignment $0.18 \pm 0.09$, null mean 0.50, percentile 0.5%, $z = -3.3$. The negation residual is *above* random (weak but correctly-aligned signal), and the NOR residual is significantly *below* random (strong but inverted signal). Neither is noise.

**Direction consistency.** We compute pairwise cosine similarity of residuals across all concepts (e.g., comparing the negation residual for "not a cat" with that for "not a dog"). Negation residuals have mean cosine $0.65 \pm 0.08$, NOR residuals have mean cosine $0.62 \pm 0.07$. A random baseline yields mean cosine $0.00 \pm 0.04$. Both operators produce highly consistent directions across concepts, confirming structured signal rather than noise.

**Concept specificity.** We shuffle residuals across concepts (assign dog's residual to cat's evaluation) and measure mean alignment over 10000 permutations. Original mean alignment: 0.65, shuffled mean: $0.51 \pm 0.01$ ($p < 0.0001$). Shuffling destroys alignment, so residuals encode concept-specific rather than generic information.

### C.2. Is Negation Encoded in the Representation?

We next ask whether negation signals exist in the text embeddings, independent of the span-residual decomposition.

**Direct analysis of the negation direction.** We analyze $\Delta_X = g(\text{"not } X\text{"}) - g(\text{"}X\text{"})$ directly. Delta magnitude: 0.31 (31% of embedding norm). Pairwise cosine across concepts: $0.62 \pm 0.09$. Delta AUC (correct negation ranking): 0.70. Mean $z_\Delta$ for images *with* concept: $-0.017$, *without*: $+0.008$. The "not" token contributes a consistent, correctly-aligned direction. Negation *is* encoded in the representation, the failure is at the interface level.

**Negation phrasing comparison.** We compare different negation phrasings for single concepts:

| Phrasing | Alignment |
|---|---|
| "not a X" | 0.65 |
| "without a X" | 0.59 |
| "lacking a X" | 0.55 |
| "no X" | 0.53 |

Different negation words contribute varying amounts of operator signal, with "not" encoding negation most strongly. Negation tokens are not ignored by the encoder.

## C.3. Why Does NOR Invert?

Finally, we investigate why NOR ($\neg A \wedge \neg B$) shows systematically inverted alignment (AUC far below chance).

**Grammar vs. semantics.** We compare "`neither A nor B`" with "`not A and not B`"—logically equivalent but grammatically distinct. "Neither...nor" alignment: $0.19 \pm 0.10$, "not...and not" alignment: $0.18 \pm 0.10$, residual cosine similarity: 0.81. The difference is statistically significant (paired $t$-test $p < 0.001$) but practically negligible ($\Delta = 0.006$). Grammar does not drive the inversion, both constructions produce nearly identical (inverted) residuals.

**Salience encoding.** We correlate NOR residual scores $z_\perp$ with image properties. Correlation with concept $A$ presence: $+0.25$, with concept $B$ presence: $+0.18$. Mean $z_\perp$ by quadrant: BOTH (0.063) > A-ONLY (0.060) > B-ONLY (0.047) > NEITHER (0.004). The NOR residual encodes "concepts are salient," assigning higher scores to images *with* the mentioned concepts. This explains the systematic inversion: the residual tracks whether concepts are visually prominent, not whether they should be absent.

## D. Span Coefficient Analysis

The span-residual decomposition expresses each compound embedding as $\mathbf{t} = \mathbf{t}_{\text{span}} + \mathbf{t}_\perp$, where $\mathbf{t}_{\text{span}} = \alpha \mathbf{t}_A + \omega \mathbf{t}_B$ is the projection onto $\text{span}(\mathbf{t}_A, \mathbf{t}_B)$. We compute the normalized weight $\alpha/(\alpha + \omega)$ for each operator across all 3,081 COCO concept pairs.

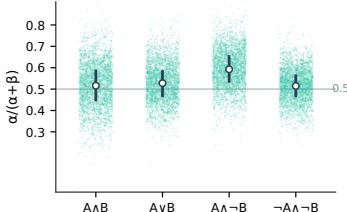

*Figure 6.* **Span coefficient distribution.** Normalized weights $\alpha/(\alpha + \omega)$ cluster around 0.5 (shaded region) for all operators, consistent with the span component approximating mean pooling. White dots indicate medians, bars show interquartile range. $A \wedge \neg B$ is slightly A-biased (0.59) because concept A is asserted while B is negated.

Figure 6 shows that all two-concept operators yield $\alpha/(\alpha + \omega) \approx 0.5$: $A \wedge B$ (0.52), $A \vee B$ (0.53), $\neg A \wedge \neg B$ (0.51). The exception $A \wedge \neg B$ (0.59) is slightly A-biased, consistent with the asymmetric phrasing "a $A$ but not a $B$" where concept A is asserted. These balanced weights confirm that the span component, which dominates holistic scores due to low operator gain, implements approximately equal-weighted averaging of atomic embeddings.

## E. Proof of Proposition 1

We prove that the pairwise inversion rate between span-only and full scoring vanishes as operator gain $G \to 0$.

**Setup.** Let $z_{\text{span}}(I) = \mathbf{t}_{\text{span}}^\top \mathbf{v}(I)$ and $z_\perp(I) = \mathbf{t}_\perp^\top \mathbf{v}(I)$ denote the span and residual score components for image $I$. The full score is $z_{\text{full}}(I) = z_{\text{span}}(I) + z_\perp(I)$. For an image pair $(I, J)$, define the pairwise differences $\Delta_s \equiv \Delta z_{\text{span}} = z_{\text{span}}(I) - z_{\text{span}}(J)$ and $\Delta_r \equiv \Delta z_\perp = z_\perp(I) - z_\perp(J)$.

**Part 1: Inversion condition.** A *rank inversion* occurs when span-only and full scoring disagree: span ranks $I > J$ (i.e., $\Delta_s > 0$) but full ranks $J > I$ (i.e., $\Delta_s + \Delta_r < 0$), or vice versa.

For span to rank $I > J$ but full to rank $J > I$:

$$\Delta_s > 0 \quad \text{and} \quad \Delta_s + \Delta_r < 0 \quad \Rightarrow \quad \Delta_r < -\Delta_s < 0. \tag{8}$$

This requires (i) opposite signs: $\text{sign}(\Delta_r) \neq \text{sign}(\Delta_s)$, and (ii) $|\Delta_r| > |\Delta_s|$. The symmetric case ($\Delta_s < 0, \Delta_s + \Delta_r > 0$) yields the same conditions. Thus, inversion requires $|\Delta_s| < |\Delta_r|$ with opposite signs. $\quad\square$

**Part 2: Inversion rate under Gaussian assumption.** We make two modeling assumptions:

1. *i.i.d. sampling*: Image pairs $(I, J)$ are drawn independently from the same distribution. This implies $\text{Var}(\Delta_s) = 2\text{Var}(z_{\text{span}})$ and $\text{Var}(\Delta_r) = 2\text{Var}(z_\perp)$.

2. *Approximate isotropy*: Image embeddings have approximately isotropic covariance, i.e., $\mathbb{E}[\mathbf{v}\mathbf{v}^\top] \approx \sigma^2 \mathbf{I}$. Combined with $\mathbf{t}_{\text{span}} \perp \mathbf{t}_\perp$, this implies $\text{Cov}(z_{\text{span}}, z_\perp) \approx 0$, so the score components are approximately uncorrelated.

*Empirical verification.* We verify assumption 2 on COCO val2017 across 500 concept pairs. The mean covariance $\text{Cov}(z_{\text{span}}, z_\perp)$ is small in absolute terms ($\sim 10^{-4}$), though the correlation coefficient is moderate (mean $|\rho| \approx 0.31$). This apparent discrepancy arises because both score components have small variance; the absolute covariance contributes little to total variance even when normalized correlation is non-negligible. The approximation introduces modest error but does not affect qualitative conclusions. The residual's contribution remains bounded by $G$, and the monotone relationship between $G$ and inversion rate holds.

Under these assumptions, $\Delta_s \sim \mathcal{N}(0, \sigma_s^2)$ and $\Delta_r \sim \mathcal{N}(0, \sigma_r^2)$ independently. Let $W = \Delta_s + \Delta_r$ be the full score difference. Then $W \sim \mathcal{N}(0, \sigma_s^2 + \sigma_r^2)$, and $(\Delta_s, W)$

*Table 10.* **Top-$k$ success and global correctness are independent.** Even 0/10 top-$k$ violations (cat/dog) coexists with substantial pairwise errors (AUC=0.85, not 1.0). Conversely, high AUC (0.92) can produce 5/10 violations (car/cow).

| Concept Pair | AUC | Top-10 Viol. |
|---|---|---|
| *Top-k appears successful:* | | |
| cat / dog | 0.85 | 0/10 |
| bowl / cat | 0.86 | 1/10 |
| dog / fire hydrant | 0.81 | 1/10 |
| *Top-k reveals failures:* | | |
| car / cow | 0.92 | 5/10 |
| fork / laptop | 0.83 | 8/10 |
| bird / vase | 0.59 | 9/10 |
| stop sign / bus | 0.60 | 10/10 |
| carrot / orange | 0.61 | 10/10 |

is jointly Gaussian with correlation

$$\rho = \frac{\mathrm{Cov}(\Delta_s, W)}{\sigma_s \sigma_W} = \frac{\sigma_s^2}{\sigma_s\sqrt{\sigma_s^2 + \sigma_r^2}} = \frac{\sigma_s}{\sqrt{\sigma_s^2 + \sigma_r^2}} = \sqrt{1-G},$$
(9)

where $G = \sigma_r^2/(\sigma_s^2 + \sigma_r^2)$ is the operator gain.

The inversion probability is $P(\mathrm{inv}) = P(\mathrm{sign}(\Delta_s) \neq \mathrm{sign}(W)) = P(\Delta_s > 0, W < 0) + P(\Delta_s < 0, W > 0)$. For bivariate Gaussian with correlation $\rho$, a standard result gives:

$$P(X > 0, Y < 0) = \frac{1}{4} - \frac{1}{2\pi}\arcsin(\rho).$$
(10)

By symmetry, $P(\mathrm{inv}) = 2P(\Delta_s > 0, W < 0)$, so:

$$P(\mathrm{inv}) = \frac{1}{2} - \frac{1}{\pi}\arcsin(\sqrt{1-G}).$$
(11)

$\square$

## F. Limitations of Top-$k$ Evaluation

Top-$k$ success can mask systematic constraint violations. Even when top results appear correct, the score distribution may assign high similarity to many violating images, which breaks thresholding and downstream chaining.

Table 10 audits AND across diverse COCO pairs, and reports both (i) pairwise separability (AUC) and (ii) top-10 violation count. Even when top-10 appears perfect (e.g., *cat/dog*: 0/10 violations), AUC can be far from 1.0, which implies substantial pairwise errors. Conversely, high AUC does not guarantee clean top-$k$ (e.g., *car/cow*: AUC 0.92 but 5/10 violations).

Figure 7 visualizes score distributions over all COCO images for two AND queries. In *cat and dog*, the extreme right tail is dominated by BOTH images. This yields few top-$k$ violations, yet substantial overlap remains. In *stop sign and bus*, overlap is severe and top-$k$ collapses entirely into violating quadrants.

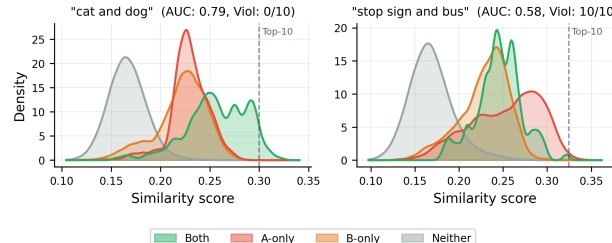

*Figure 7.* **Top-$k$ success is accidental, not semantic.** Score distributions for two AND queries. *Left:* "cat and dog" appears to work because BOTH images dominate the extreme tail, yet substantial overlap means many pairwise errors. *Right:* "stop sign and bus" fails completely because violating quadrants dominate the tail.

*Table 11.* **Aggregation method ablation.** Power mean provides the best balance between FACTOR-Bench accuracy and NegBench performance. Softer $\gamma$ values improve NegBench at the cost of FACTOR-Bench.

| Aggregation | FB Acc | MCQ Acc | Ret R@5 | Ret $\rho$ |
|---|---|---|---|---|
| *Aggregation type:* | | | | |
| Minmax | 87.5 | 56.1 | 48.5 | 0.999 |
| Probabilistic | 86.8 | 47.8 | 51.2 | 0.999 |
| Power mean ($\gamma_{\mathrm{and}}{=}{-}1$) | **85.5** | **60.3** | **50.8** | **0.999** |
| *Power mean $\gamma$ sweep:* | | | | |
| Strict ($\gamma_{\mathrm{and}}{=}{-}10$) | 86.6 | 57.5 | 48.9 | 0.999 |
| Default ($\gamma_{\mathrm{and}}{=}{-}1$) | 85.5 | 60.3 | 50.8 | 0.999 |
| Soft ($\gamma_{\mathrm{and}}{=}{-}0.5$) | 84.9 | 60.7 | 51.1 | 1.000 |
| Mean-biased ($\gamma_{\mathrm{and}}{=}0.5$) | 83.4 | 61.7 | 51.6 | 1.000 |

## G. Aggregation Method Ablation

We compare different aggregation functions for combining concept scores in LCSE. Three approaches are tested:

- **Minmax**: AND $= \min(p_i)$, OR $= \max(p_i)$
- **Probabilistic**: AND $= \prod p_i$, OR $= 1 - \prod(1 - p_i)$
- **Power mean**: $M_\gamma(\mathbf{x}) = \left(\frac{1}{n}\sum x_i^\gamma\right)^{1/\gamma}$ with separate $\gamma$ for AND/OR

Table 11 shows results on CLIP ViT-B/32 with LCSE. Power mean with $\gamma_{\mathrm{and}}{=}{-}1$, $\gamma_{\mathrm{or}}{=}10$ achieves the best balance: high FACTOR-Bench accuracy (85.5%) with strong NegBench performance. Minmax achieves highest FACTOR-Bench (87.5%) but lags on NegBench MCQ ($-4$pp). Probabilistic underperforms on both benchmarks.

The $\gamma$-sweep reveals a trade-off: stricter $\gamma$ (more min-like) improves FACTOR-Bench accuracy on synthetic operator tests, while softer $\gamma$ (closer to mean) improves NegBench performance on natural language queries. We select $\gamma_{\mathrm{and}}{=}{-}1$ as the default to balance both objectives. All power mean configurations achieve retention $\rho > 0.99$, which confirms that aggregation choice does not affect holistic signal preservation.

*Table 12.* **LCSE vs. factored-only variants.** Routing preserves retention but degrades MCQ; full replacement degrades retention. LCSE avoids both failure modes.

| Benchmark | LCSE | Routed | Full |
|---|---|---|---|
| FACTOR-Bench Acc. (%) | 85.5 | 88.9 | 89.4 |
| NegBench MCQ Acc. (%) | **60.3** | 51.8 | 56.4 |
| NegBench Retrieval R@5 (%) | **50.8** | 34.5 | 34.5 |
| Retention $\rho$ | **0.999** | 0.994 | 0.870 |

*Table 13.* **LCSE scales to 3–4 concept compound operators.** On queries with mixed polarity across multiple concepts, LCSE achieves 95.2% overall accuracy compared to 73.8% for holistic scoring.

| Concepts | $n$ | Holistic | LCSE | $\Delta$ |
|---|---|---|---|---|
| 3 | 97 | 69.1% | 93.8% | +24.7 |
| 4 | 48 | 83.3% | 97.9% | +14.6 |
| Overall | 145 | 73.8% | 95.2% | +21.4 |

## H. LCSE vs. Factored-Only

A natural question is whether the LCSE correction is necessary, or whether factored scoring alone suffices. *Factored-only* replaces the holistic signal with $p_{\text{logic}}$ and discards holistic information (scene context, spatial layout, stylistic cues) that $p_{\text{logic}}$ cannot capture. We evaluate two variants: *routed*, which uses $s_{\text{logic}} = \text{logit}(p_{\text{logic}})/\beta + \mu$ for operator queries and $s_{\text{hol}}$ otherwise, and *full*, which uses $p_{\text{logic}}$ for all queries.

Table 12 compares all three methods (CLIP ViT-B/32, power-mean aggregation). Both factored-only variants outperform LCSE on FACTOR-Bench (+3–4pp), where synthetic tests do not benefit from holistic context. However, each variant fails differently on natural-language benchmarks. Routing preserves retention ($\rho = 0.994$, since only 2.5% of COCO captions contain operators) but degrades MCQ to 51.8% because it mixes $s_{\text{logic}}$ and $s_{\text{hol}}$ within the same comparison set. Full replacement avoids this mixing but destroys retention ($\rho = 0.87$) by discarding holistic scores for all queries. LCSE's additive correction avoids both failure modes.

## I. Multi-Concept Compound Operators

The main experiments evaluate LCSE on two-concept queries. Here we demonstrate that LCSE scales naturally to three or more concepts with compound Boolean operators (e.g., "a cat and a dog but no bird"). The power mean aggregation (Eq. 6) operates on arbitrary-length concept lists, so no architectural changes are required.

We evaluate on FACTOR-Bench COMPOUND tests, which contrast queries with 3–4 concepts and mixed polarity. For example, 3-concept tests compare "a $A$ and a $B$ and no $C$" vs. "no $A$ and a $B$ and a $C$", while 4-concept tests compare "a $A$ and a $B$ and no $C$ and no $D$" vs. "no $A$ and no $B$ and a $C$ and a $D$". Both captions contain the same concepts but differ in polarity assignment.

Table 13 shows that LCSE achieves 95.2% overall accuracy on compound operators with 3–4 concepts, compared to 73.8% for holistic scoring (+21.4pp). The improvement is consistent across both concept counts, which shows that power mean aggregation scales naturally to more concepts

without modification.

**Nested Boolean structure.** The compound queries above are *flat* conjunctions over mixed-polarity concepts. Factored inference also handles *nested* structure by aggregating recursively over the parse tree. Each subtree is reduced with its operator's power mean, and the result feeds its parent. We test this on 3-concept nested queries that share concepts but differ in grouping (e.g., "$(A \wedge B) \vee C$" vs. "$(A \vee B) \wedge C$"), constructed following the FACTOR-Bench protocol (OWL-ViT validated, position randomized). Holistic scoring is near chance (51.0%), as nesting is invisible to bag-of-concepts pooling, while nested LCSE reaches 80.5% (+29.5pp), comparable to its flat-operator gain (+27.2pp). The only change is recursive aggregation; concept scoring, calibration, and the correction are unchanged.

## J. Logical Equivalence Analysis

A principled operator mechanism should satisfy basic logical equivalences: queries that are logically equivalent should produce identical scores. We test three equivalences on FACTOR-Bench:

- De Morgan's laws: "neither $A$ nor $B$" $\equiv$ "not $A$ and not $B$"
- Double negation: "not without $A$" $\equiv$ "$A$"
- Commutativity: "$A$ and $B$" $\equiv$ "$B$ and $A$"

Figure 8 reports violation rates (scores differ by $> 0.001$ in cosine similarity) across methods. A pure **Factored** approach achieves 0% violations by construction: it extracts atomic concept scores and applies exact Boolean formulas, so that equivalent expressions lead to identical results.

Holistic models violate equivalences 84–100% of the time because different text phrasings produce different embeddings. Fine-tuned variants often have *higher* violation rates than their base models (e.g., NegCLIP 93% vs. CLIP 90%), which suggests that negation-focused training can disrupt other equivalences.

LCSE produces *identical* violation rates to its holistic backbone (LCSE-CLIP = CLIP = 90%, LCSE-SigLIP 2 = SigLIP 2 = 88%). This is expected: the FACTOR-Bench oracle parser canonicalizes equivalent forms to the same



*Figure 8.* **Logical equivalence violations.** Violation rate (%) when logically equivalent queries produce different scores ($> 0.001$ threshold in cosine similarity). Factored scoring achieves 0% by construction. LCSE matches its holistic backbone exactly, since the oracle parser canonicalizes equivalent forms.

*Table 14.* **Retrieval-time scoring overhead** (RTX 4090, CLIP ViT-B/32, ms per query, median). LCSE adds $\sim 4\%$ over holistic across pool sizes with pre-cached concept embeddings and a fused scoring step.

| Pool size | Holistic (ms) | LCSE (ms) | Overhead |
|---|---|---|---|
| 5,000 | 1.73 | 1.81 | +4.6% |
| 50,000 | 1.84 | 1.92 | +4.3% |
| 500,000 | 3.05 | 3.18 | +4.3% |

logical structure, so the additive correction is identical for both queries and cancels exactly. The gap to Factored (0%) represents an opportunity for future methods that prioritize logical consistency.

# K. Retrieval-Time Overhead

We measure the wall-clock cost of LCSE scoring on an RTX 4090 (CLIP ViT-B/32, precomputed image embeddings, pre-cached concept embeddings, median of 50 runs over 20 queries; parser inference is a one-time per-query cost, independent of pool size, and is excluded). LCSE fuses the holistic and concept similarities into a single matrix multiply over the image pool, so its per-image cost is independent of concept count ($m=1$ and $m=2$ differ by $<0.01$ms). Table 14 reports $\sim 4\%$ overhead across pool sizes up to 500K images. This figure refers to the fused path; a naive implementation that scores each concept separately incurs larger overhead (up to $\sim 1.4\times$ at 500K images), which the fused formulation avoids.

