# OpenReview forum: "Similarity Is Not Logic: Factored Inference for Dual-Encoder Vision-Language Models"
_ICML.cc/2026/Conference — ICML 2026 regular_

### Official Review · Reviewer_gND1 · 2026-03-12

**Soundness:** 3
**Presentation:** 3
**Significance:** 3
**Originality:** 3
**Overall Recommendation:** 4
**Confidence:** 3

**Summary:**

This paper studies a fundamental limitation of dual-encoder vision–language models such as CLIP and SigLIP when handling compositional queries involving logical operators. The authors show that similarity-based retrieval behaves like a bag-of-concepts interface, where scores effectively average evidence for individual concepts and therefore fail to correctly execute logical constraints such as negation or conjunction. To address this, the paper proposes a factored inference framework that separates concept evidence extraction from constraint execution, and introduces a training-free method called Logic-Constrained Score Editing (LCSE) that adjusts similarity scores by explicitly applying logical operators over concept-level evidence. Experiments on the proposed FACTOR-Bench and existing negation benchmarks demonstrate that LCSE substantially improves logical constraint satisfaction while largely preserving standard retrieval performance, suggesting that many compositional failures of current VLMs arise from the similarity interface rather than the underlying representations.

**Compliance With Llm Reviewing Policy:**

Affirmed.

**Key Questions For Authors:**

1. Limited complexity of compositional logic. The benchmark mainly focuses on relatively simple Boolean compositions (e.g., conjunction, disjunction, and negation) over a small number of visual concepts. While this controlled setting is useful for analyzing the behavior of the proposed method, it does not fully demonstrate how well the approach performs under more complex compositional scenarios. For example, queries involving multiple interacting concepts, nested logical structures, spatial relations, or attribute bindings are not explored. As a result, it remains unclear whether the proposed method can generalize effectively to more realistic and complex reasoning settings.

2. Lack of analysis on incorrect logical inference. Although the paper demonstrates overall performance improvements when logical constraints are correctly applied, it does not analyze how the system behaves when logical reasoning is incorrect or when the parsing of logical structures fails. Since the method relies on extracting logical structures and concept-level evidence, errors in logical interpretation or concept detection could significantly affect the final ranking. A more detailed analysis showing how performance changes under incorrect logical predictions, or across different logical operators, would provide a better understanding of the robustness of the proposed approach.

**Limitations:**

see in questions

**Strengths And Weaknesses:**

Soundness

The paper is generally technically sound. The authors provide a clear empirical and mechanistic analysis showing that dual-encoder VLM similarity scoring behaves like a bag-of-concepts aggregation, which fails to properly execute logical operators. The proposed method, LCSE, is conceptually simple and well motivated by the analysis. The experimental evaluation is reasonably designed and includes multiple benchmarks, such as FACTOR-Bench and NegBench, as well as retrieval retention tests to verify that the proposed correction does not significantly degrade standard retrieval performance. The empirical results support the main claims of the paper. That said, the evaluation is limited to relatively simple Boolean operators and a small set of concepts, and the method relies on accurate parsing of logical structure and reliable concept detection by the underlying VLM. As a result, while the approach appears valid for the studied setting, its robustness and generalization to more complex compositional queries remain uncertain.

Presentation

The paper is generally well written and clearly structured. The motivation is well articulated, and the mechanistic analysis of the failure mode of similarity-based retrieval provides a useful conceptual framing for the proposed method. The presentation of the factored inference framework and LCSE is relatively straightforward and easy to follow. The experiments are organized clearly, and the figures illustrating the operator-insensitive behavior of similarity scores are helpful. However, a few aspects of the presentation could be improved. In particular, the discussion of related work on compositional reasoning and interface-level modifications to VLM retrieval could be expanded to more clearly position the work relative to existing approaches. Additionally, the paper could benefit from more discussion on limitations and potential failure cases, especially for complex language queries involving nested logical structures or relational reasoning.

Significance

The paper addresses an important limitation of current vision-language retrieval models. As dual-encoder architectures such as CLIP and SigLIP are widely used in retrieval systems and multimodal pipelines, understanding their limitations in compositional reasoning is highly relevant. By highlighting that the core issue lies in the similarity interface rather than the underlying representation, the work provides a useful perspective for future research on multimodal retrieval and reasoning. The proposed method also has practical appeal since it is training-free and can be applied to existing models without modifying their architecture. However, the scope of the contribution is somewhat limited to logical constraints over visual concepts, and it remains unclear how well the approach would extend to more complex reasoning tasks such as relational reasoning, attribute binding, or counting. As such, while the work offers useful insights and practical improvements for a specific class of queries, its broader impact will depend on whether the proposed framework can generalize to more complex compositional settings.

Originality

The paper offers a useful perspective on the limitations of similarity-based retrieval in dual-encoder vision–language models. While the general idea of separating perception from structured reasoning has been explored in prior work on compositional reasoning and neuro-symbolic methods, this paper provides a focused analysis showing that many logical failures arise specifically from the similarity interface rather than from deficiencies in the underlying representations. This insight helps clarify an important failure mode of widely used models such as CLIP and SigLIP. The proposed LCSE method is relatively simple and operates as an inference-time score adjustment rather than introducing a new model architecture or training objective. As a result, the methodological novelty is moderate. Nevertheless, the work combines mechanistic analysis with a practical inference-time correction that can be applied to existing models without retraining, which provides a useful and practical contribution to understanding and improving compositional behavior in vision–language retrieval systems.

Weakness

Limited complexity of compositional logic.
The benchmark mainly focuses on relatively simple Boolean compositions (e.g., conjunction, disjunction, and negation) over a small number of visual concepts. While this controlled setting is useful for analyzing the behavior of the proposed method, it does not fully demonstrate how well the approach performs under more complex compositional scenarios. For example, queries involving multiple interacting concepts, nested logical structures, spatial relations, or attribute bindings are not explored. As a result, it remains unclear whether the proposed method can generalize effectively to more realistic and complex reasoning settings.

Lack of analysis on incorrect logical inference.
Although the paper demonstrates overall performance improvements when logical constraints are correctly applied, it does not analyze how the system behaves when logical reasoning is incorrect or when the parsing of logical structures fails. Since the method relies on extracting logical structures and concept-level evidence, errors in logical interpretation or concept detection could significantly affect the final ranking. A more detailed analysis showing how performance changes under incorrect logical predictions, or across different logical operators, would provide a better understanding of the robustness of the proposed approach.

---

> ### Author Rebuttal · Authors · 2026-03-30
>
> We thank the reviewer for the thorough assessment. The two concerns raised (compositional scope and failure analysis) are both important, and we address each below.
>
> **Limited compositional scope (W1, Q1).** We agree that the paper is intentionally scoped to Boolean-style compositional retrieval over visual concepts, where correctness is unambiguous and we can cleanly separate execution failures from representation failures. We will revise the framing to make this scope more explicit. Within our scope, Appendix I (Multi-Concept Compound Operators) evaluates **3-4 concept mixed-polarity compound queries**, and it shows LCSE improves over holistic scoring from 73.8% to **95.2%** (+21.4pp).
>
> To go further and directly test **nested logical structures**, we constructed 200 pairwise samples following FACTOR-Bench's construction methodology with 3-concept nested queries (e.g., "(A AND B) OR C" vs "(A OR B) AND C") and evaluated using oracle parses with recursive power-mean aggregation over parse trees. Holistic scoring drops to 51.0% on these queries (near chance), while nested LCSE achieves 80.5% **(+29.5pp)**, which is comparable to the flat FACTOR-Bench gain (+27.2pp). The only change required was recursive aggregation, while concept scoring, calibration, and the LCSE correction are unchanged.
>
> Attribute binding, spatial relations, and counting involve different failure mechanisms from the Boolean aggregation mismatch studied here. Factored inference (decompose, extract, execute) may extend to these settings, though it would depend on the backbone providing reliable evidence for each structure type.
>
> **Failure analysis (W2, Q2).** We evaluate under two conditions: (1) **oracle-parsed FACTOR-Bench**, which isolates the interface-level execution problem from parsing noise; and (2) **end-to-end evaluation with an LLM parser** on NegBench and COCO retrieval captions. To quantify the parser's impact, we re-ran FACTOR-Bench with GPT-4.1-mini parses. Performance changes only slightly (85.5% → 85.2%), with 0 wrong-sign corrections. See our response to Reviewer 6cvT for the full per-operator breakdown.
>
> To provide a more detailed failure analysis, we stratified FACTOR-Bench by concept-detection quality. For each of 77 COCO categories, we computed per-concept detection AUC (how well CLIP similarity separates images with vs. without that concept, with mean AUC = 0.885). For each sample, we took the minimum AUC across involved concepts (the "weakest link," since LCSE uses all concept scores in the correction):
>
> | Concept AUC bin | n | Holistic (%) | LCSE (%) | Δ (pp) |
> |-----------------|---|-------------|---------|--------|
> | High (>0.90) | 417 | 57.8 | 90.4 | +32.6 |
> | Medium (0.75–0.90) | 479 | 61.4 | 85.2 | +23.8 |
> | Low (<0.75) | 204 | 52.0 | 76.0 | +24.0 |
>
> LCSE exceeds holistic in every bin. When detection is strong, LCSE reaches **90.4%**, which suggests that the remaining errors in this regime are better explained by evidence quality than by operator execution. When detection is weak, performance drops predictably to 76.0%, but still exceeds holistic by +24pp, which is consistent with the paper's claim that LCSE **shifts failures from execution to evidence extraction**.
>
> In the revision, we will make the scope and limitations more explicit, expand the related-work discussion, and add the new experimental results.

---

> > ### Author Rebuttal · Reviewer_gND1 · 2026-04-08
> >
> > Thank you for the author's reply. I have no more questions now and will keep the score.

---

### Official Review · Reviewer_P54B · 2026-03-12

**Soundness:** 3
**Presentation:** 4
**Significance:** 3
**Originality:** 4
**Overall Recommendation:** 5
**Confidence:** 3

**Summary:**

This paper investigates a fundamental limitation of dual-encoder vision–language models, particularly those that rely on similarity-based retrieval interfaces such as CLIP and SigLIP. The authors show that although these models perform well on standard zero-shot retrieval tasks, they often struggle with compositional queries involving logical operators such as conjunction and negation. The paper argues that this issue arises from the nature of similarity-based scoring, which tends to aggregate evidence for individual concepts rather than strictly enforcing logical constraints between them. To systematically analyze this limitation, the authors introduce FACTORBench, a benchmark specifically designed to evaluate how well models handle compositional and constraint-based retrieval queries.

To address this challenge, the authors propose a framework called factored inference, which separates concept evidence extraction from logical constraint execution. Based on this idea, they introduce a training-free method called Logic-Constrained Score Editing (LCSE), which modifies similarity scores using concept-level signals derived from frozen encoders. This approach allows logical constraints to be enforced without retraining the underlying model and without degrading standard retrieval performance. Experimental results demonstrate that LCSE substantially improves performance on compositional retrieval tasks, outperforming several fine-tuned baselines and achieving strong gains on FACTORBench as well as other negation-focused benchmarks.

Overall, the paper is clearly written, and the proposed method demonstrates promising and competitive results. I appreciate the problem formulation and its potential practical applications, and based on the overall assessment, I would recommend accepting the paper.

**Compliance With Llm Reviewing Policy:**

Affirmed.

**Final Justification:**

Thank you to the authors for the thoughtful rebuttal and for clarifying several aspects of the work. I appreciate the effort to address the concerns raised. While some points have been clarified, a few of my original concerns remain only partially resolved. Given that my initial score was already relatively favorable, I will maintain my current evaluation.

**Key Questions For Authors:**

One key question for the authors concerns the practical robustness and scalability of the proposed framework, particularly the reliance on query decomposition into logical atoms before applying LCSE. The current setup appears to assume an oracle-style parser for breaking queries into components, which demonstrates the method’s upper-bound potential; however, it would be helpful to understand how performance degrades when this step is handled by real-world large language models, especially for ambiguous or linguistically complex natural language queries. In addition, since the evaluation relies on the proposed FACTORBench, it would be useful to know how well the observed improvements translate to more open-ended or natural queries encountered in practical retrieval systems. Another related concern is the computational overhead introduced by the factored inference approach: because LCSE requires multiple atomic similarity evaluations when applied to models such as CLIP and SigLIP, it would be valuable for the authors to clarify the latency implications when scaling to large-scale, real-time retrieval environments. Finally, it would be interesting to know whether there are scenarios in which the holistic global representations of standard vision–language models might outperform factored logical reasoning, and whether a dynamic strategy could be developed to decide when LCSE should be applied.

**Limitations:**

The primary limitations of this work stem from its reliance on an idealized query decomposition process and the potential overhead introduced by its inference strategy. In particular, the proposed LCSE relies on a factored approach that requires complex queries to be decomposed into atomic concepts before applying score editing. While the paper assumes an oracle-style parser to demonstrate the method’s upper-bound performance, real-world applications would likely rely on automated large language model–based parsing, which may introduce errors or ambiguities when handling complex natural language queries. Additionally, although the evaluation benchmark FACTORBench provides a controlled setting to test logical operators such as conjunction and negation, it focuses on a relatively limited set of compositional patterns and may not fully capture the diversity of natural language queries encountered in practical retrieval scenarios. Another concern is the computational overhead of the factored inference approach: because LCSE requires multiple atomic similarity evaluations when applied to models such as CLIP and SigLIP, it may introduce additional latency that could be challenging for large-scale, real-time retrieval systems. Finally, while the method effectively enforces Boolean constraints, decomposing queries into atomic components may overlook certain holistic contextual signals or global semantic relationships that standard dual-encoder retrieval models naturally capture.

**Strengths And Weaknesses:**

I think the strongest part of this paper is how it identifies a really specific, structural 'Bag-of-Concepts' failure in dual-encoders that many of us suspected but haven't seen mapped out this clearly. The mechanistic analysis is great because it shows that the model does actually have the signals for things like negation in its embeddings, but the dot-product interface just completely washes them out. I also really like that LCSE is a training-free, 'plug-and-play' solution; being able to fix logical failures without expensive fine-tuning is a huge win for accessibility. The FACTOR-Bench dataset they built is another high point—it’s a much-needed, clean way to test these specific logical constraints. Finally, seeing the method generalize so well across different backbones like CLIP and SigLIP makes the results feel very dependable and widely applicable.

On the flip side, there are a few things that gave me pause. First, the method appears to rely heavily on an oracle-style parser to decompose queries into logical components, and it would be useful to understand how robust the approach is when this step is replaced with a practical large language model–based parser that may introduce errors. Second, the factored inference approach may introduce an inherent bias: by decomposing queries into atomic components, the method might lose certain holistic or contextual cues that standard similarity-based retrieval sometimes captures effectively. Finally, while the zero-shot improvements are impressive, the paper provides limited discussion about the computational overhead of executing multiple atomic queries during retrieval. In large-scale production environments, this could potentially introduce latency or efficiency trade-offs that are not fully explored in the current evaluation.

---

> ### Author Rebuttal · Authors · 2026-03-30
>
> We thank the reviewer for the thoughtful and encouraging assessment. We are glad that the Bag-of-Concepts diagnosis, the training-free design, FACTOR-Bench, and the cross-backbone generalization each came through as intended.
>
> **Parser robustness (W1).** The paper evaluates two regimes. FACTOR-Bench uses **oracle parses by design** to isolate execution from parsing noise, while NegBench and retention tests use end-to-end **GPT-4.1-mini parses**. To quantify the impact of real parsing on our controlled benchmark, we re-ran FACTOR-Bench with LLM parses replacing the oracle:
>
> | Operator | LCSE Oracle (%) | LCSE LLM (%) | Oracle->LLM (pp) |
> |----------|----------------:|-------------:|----------------:|
> | NOT      | 88.0            | 88.0         | 0.0             |
> | AND      | 84.7            | 82.7         | -2.0            |
> | OR       | 90.7            | 90.7         | 0.0             |
> | BUT NOT  | 82.8            | 82.8         | 0.0             |
> | NEITHER  | 82.4            | 82.4         | 0.0             |
> | **Overall** | **85.5**     | **85.2**     | **-0.3**        |
>
> Four of five operators show zero change, and no parse error causes LCSE to apply an incorrect operator. Beyond this controlled setting, the NegBench results in Table 2 (60.3% MCQ, end-to-end LLM parses) provide evidence on more natural, LLM-rephrased captions. We agree that this does not yet establish robustness to fully open-ended user queries, and will make that boundary explicit.
>
> **Holistic signal preservation (W2, Q).** This concern is exactly why we introduced **LCSE rather than factored scoring alone**. Appendix H tests two factored-only alternatives that illustrate the risk: (1) a **routing** variant that switches between factored scores (for operator queries) and holistic scores (otherwise), which mixes two incomparable score scales and degrades NegBench MCQ to 51.8%; (2) a **full replacement** variant that uses factored scores for all queries, which discards scene context and collapses retention to ρ = 0.87. LCSE avoids both failure modes. Its additive correction preserves the holistic signal and adjusts only the aggregation mismatch, and achieves ρ = 0.999 on both COCO and PASCAL (Table 2). We will bring this motivation forward in the main text.
>
> **Overhead (W3).** We quantify retrieval-time scoring overhead on an RTX 4090 (precomputed image embeddings, pre-cached concept embeddings, median of 50 runs over 20 queries):
>
> | Pool size | Holistic (ms) | LCSE (ms) |
> |-----------|--------------|-----------|
> | 5,000 | 1.73 | 1.81 |
> | 50,000 | 1.84 | 1.92 |
> | 500,000 | 3.05 | 3.18 |
>
> LCSE adds **~4% overhead** across all pool sizes tested, including 500K images.
>
> **Dynamic strategy (Q).** This is an interesting suggestion. The current method already behaves like a simple dynamic strategy in one important case, where for the 97.5% of COCO captions without explicit Boolean operators or negation, the LCSE correction is exactly zero, so the score reduces to the holistic baseline. That is, LCSE only intervenes when the parsed query indicates that operator-aware correction is needed. A more adaptive trigger, for example based on parse confidence or uncertainty, would be a natural direction for future work.
>
> We will add the end-to-end parser result and timing table to the main text, and surface the holistic-preservation ablation (currently in the appendix) in the main discussion.

---

> > ### Author Rebuttal · Reviewer_P54B · 2026-04-03
> >
> > Thank you to the authors for the thoughtful rebuttal and for clarifying several aspects of the work. I appreciate the effort to address the concerns raised. While some points have been clarified, a few of my original concerns remain only partially resolved. Given that my initial score was already relatively favorable, I will maintain my current evaluation.

---

> > > ### Author Response · Authors · 2026-04-06
> > >
> > > Thank you for the supportive review and for the engagement throughout the discussion.

---

### Official Review · Reviewer_6cvT · 2026-03-13

**Soundness:** 3
**Presentation:** 3
**Significance:** 3
**Originality:** 3
**Overall Recommendation:** 4
**Confidence:** 3

**Summary:**

This paper studies the limitations of similarity-based retrieval in dual-encoder vision–language models and shows that queries containing logical operators often fail because cosine similarity aggregates concept evidence in a “bag-of-concepts” manner and does not properly enforce logical constraints. To address this, the paper proposes factored inference, which separates concept evidence extraction from logical constraint execution. Building on this idea, the authors introduce LCSE (Logic-Constrained Score Editing), a training-free inference method that edits similarity scores using concept-level evidence to enforce logical operators. They also introduce FACTOR-Bench, a benchmark for evaluating logical compositionality in retrieval. Experiments show that LCSE substantially improves performance on FACTOR-Bench and NegBench.

**Compliance With Llm Reviewing Policy:**

Affirmed.

**Final Justification:**

**Soundness.** The paper provides a clear analysis of failures in similarity-based retrieval for logical queries and supports its claims with strong empirical results on FACTOR-Bench and NegBench. The evidence is convincing for the studied settings.

**Originality.** The work introduces a new framing of "bag-of-concepts" failures and proposes LCSE, a simple yet effective training-free method for enforcing logical constraints. FACTOR-Bench is also a valuable new benchmark.

**Significance.** The paper addresses an important issue in vision–language retrieval and offers meaningful insights. However, as mentioned in the Limitations section, it cannot solve nested or ambiguous queries, and the extension would require nontrivial modification on top of the method.

**Clarity.** The paper is well-written and well-structured.

Regarding my concerns:

W1: Concern on the scalability to larger pools and computational overhead is addressed by the additional experiments.

W2: Concern on the parameter sensitivity and calibration is resolved by additional results.

W3: Concern on the parser robustness is addressed by the explanation in the rebuttal.

Overall, the work makes a useful contribution despite some limitations.

Final recommendation: I support weak acceptance.

**Key Questions For Authors:**

1. It would be useful to discuss the calibration and inference overhead, and how we can apply it more efficiently to a large image or query pool.
2. Regarding the generalization of the calibration, how sensitive are the hyperparameters $\mu$ and $\beta$ to different image and query distributions (e.g., medical images)?

**Limitations:**

yes

**Strengths And Weaknesses:**

Strengths:
1. The problem is well-motivated and clearly stated in Section 3. This section also provides a novel and rigorous analysis of the problem and the cause.
2. The improvements on FACTOR-Bench and NegBench are significant, verifying the effectiveness of this method. The comparison of the prior baselines is thorough.
3. The research is presented in a clear and ordered manner.

Weaknesses:
1. It is unclear whether this method can be applied efficiently to large image pools or query pools, which is common in retrieval settings.
2. The effectiveness of LCSE relies on the sigmoid parameters $\mu$ and $\beta$, especially $\mu$ based on Table 6 and 7 (performance drops by 17% after changing $\mu$ from 0.22 to 0.18). Also, the paper shows that $\mu$ is highly model-dependent ($\mu=0.22$ for CLIP vs. $\mu=0.05$ for SigLIP 2). If these parameters are not perfectly tuned for a specific distribution, the negation operator could fail to flip the sign of the correction, potentially leading to worse-than-holistic performance.
3. The method relies on an LLM parser to convert text into a logical schema. It is unclear how ambiguity of the query or the failure of the parser affects the retrieval performance.

---

> ### Author Rebuttal · Authors · 2026-03-30
>
> We thank the reviewer for the review, and for recognizing the strength of the Section 3 analysis and the empirical gains. The three concerns raised (scalability, calibration, and parser robustness) are all practical and important. We address each with new quantitative evidence.
>
> **Scalability (W1, Q1).** We quantify retrieval-time scoring overhead on an RTX 4090 (CLIP ViT-B/32, precomputed image embeddings, pre-cached concept embeddings, median of 50 runs over 20 queries, excluding parser inference, which is a one-time per-query cost independent of pool size).
>
> | Pool size | Holistic (ms) | LCSE (ms) |
> |-----------|--------------|-----------|
> | 5,000 | 1.73 | 1.81 |
> | 50,000 | 1.84 | 1.92 |
> | 500,000 | 3.05 | 3.18 |
>
> LCSE adds **~4% overhead** across all pool sizes tested, including 500K images. The scoring step fuses holistic and concept similarities into a single matrix multiply over the image pool, so the per-image cost is independent of concept count (m=1 and m=2 differ by <0.01ms in our tests).
>
>
> **μ sensitivity (W2, Q2).** The full calibration ablation (Appendix A):
>
> | μ | Overall Accuracy (%) | vs. Best Baseline (73.2%) |
> |---|---------------------|--------------------------|
> | 0.18 | 68.5 | -4.7pp |
> | 0.20 | 81.1 | **+7.9pp** |
> | **0.22** | **85.5** | **+12.3pp** |
> | 0.24 | 83.4 | **+10.2pp** |
> | 0.26 | 78.1 | **+4.9pp** |
>
> The steep drop occurs only at the most extreme tested value (μ=0.18). Within μ in [0.20, 0.26], all values remain above the best baseline by +4.9pp to +12.3pp. LCSE exceeds holistic CLIP (58.3%) at every tested μ, including the extreme μ=0.18 (68.5%, +10.2pp).
>
> Regarding model-dependence: μ approximates the concept-presence decision boundary, so it naturally differs across backbones (μ=0.22 for CLIP, μ=0.05 for SigLIP 2), just as any classifier needs its own threshold. To quantify calibration cost, we tested how much held-out data is needed to recover this threshold (5 random seeds per size, grid search over 7 μ values). **50 held-out images suffice** to recover μ=0.22 exactly for CLIP (zero variance for N in {50, 100, 200}). SigLIP 2 converges to 0.055 (vs. paper optimum 0.05, ~1.2pp accuracy difference). The full calibration protocol is a grid search over μ in {0.16–0.28, step 0.02} and β in {10–60, step 10}, and it completes in ~2s on a single GPU. β varies by at most ~1% across [30, 60] (Tables 6 and 7 of Appendix A).
>
> **On sensitivity to different distributions (Q2):** the same μ=0.22, β=30 is used without recalibration across all benchmarks in Table 2, including VOC2007 and PASCAL. For a new domain or backbone, recalibration follows the same protocol on a small held-out set, though LCSE also depends on the backbone providing reliable concept evidence in that domain.
>
> **Parser robustness (W3).** When the parser misses an operator, LCSE defaults to holistic scoring rather than applying an incorrect correction. This is because missed operators cause the correction term to vanish exactly ($p_{\text{logic}} = p_{\text{soft}}$). The same fallback applies to genuinely ambiguous queries where no clear Boolean structure is present. All NegBench and retention results in Table 2 already use real **GPT-4.1-mini** parses (the reported **60.3% NegBench MCQ** and **ρ = 0.999 retention** are end-to-end results, not oracle conditions).
>
> We used  oracle parses by design in FACTOR-Bench to isolate the scoring interface from parser noise. However, to quantify the oracle-to-LLM gap, we re-ran the FACTOR-Bench operator split (1,100 pairwise samples) with GPT-4.1-mini parses:
>
> | Operator | Holistic CLIP (%) | LCSE Oracle (%) | LCSE LLM (%) | Oracle->LLM (pp) |
> |----------|------------------:|----------------:|-------------:|----------------:|
> | NOT      | 58.0              | 88.0            | 88.0         | 0.0             |
> | AND      | 68.7              | 84.7            | 82.7         | -2.0            |
> | OR       | 42.0              | 90.7            | 90.7         | 0.0             |
> | BUT NOT  | 66.0              | 82.8            | 82.8         | 0.0             |
> | NEITHER  | 54.4              | 82.4            | 82.4         | 0.0             |
> | **Overall** | **58.3**       | **85.5**        | **85.2**     | **-0.3**        |
>
>
> The oracle-to-LLM gap is 0.3pp overall, with four of five operators unchanged. The entire gap comes from 18 misparsed captions with only 3 changed answers out of 1,100 samples, and for each case the correction drops out and LCSE defaults to holistic scoring. The remaining errors are better explained by concept-detection quality than by parsing or execution, and our response to reviewer gND1 quantifies this with a per-concept stratification.
>
> We will add the cross-domain limitation, timing results, and parser robustness analysis to the revision.

---

> > ### Author Rebuttal · Reviewer_6cvT · 2026-04-04
> >
> > Thank you for the detailed response and additional results which addressed my concerns. I do not have further questions and will raise the score to 4.

---

> > > ### Author Response · Authors · 2026-04-06
> > >
> > > Thank you for confirming your concerns are resolved, and we appreciate the constructive discussion. We noticed the numerical score on OpenReview hasn't updated yet, and just wanted to mention it in case it didn't go through the system.

---

### Official Review · Reviewer_dm4j · 2026-03-13

**Soundness:** 3
**Presentation:** 3
**Significance:** 3
**Originality:** 3
**Overall Recommendation:** 4
**Confidence:** 2

**Summary:**

This paper studies ranking failures of dual-encoder VLMs on Boolean-style compositional queries and argues that the main issue lies in how the similarity interface aggregates evidence, rather than in a total absence of operator-related signal. It proposes a training-free factored inference method, LCSE, and supports the claim with FACTOR-Bench, NegBench, and retrieval-retention experiments.

**Compliance With Llm Reviewing Policy:**

Affirmed.

**Final Justification:**

I don't have further questions

**Key Questions For Authors:**

As mentioned in the Weakness.

**Limitations:**

Yes

**Strengths And Weaknesses:**

Strengths:
The paper asks a clear question, and the empirical argument is well-layered.

Weakness:
1. The main argument is that operator-related signal exists in the representation but is not properly executed by scalar similarity at ranking time. This is plausible, but the paper could still make a tighter case for why the interface should be viewed as the primary bottleneck rather than one factor among several coupled ones.
2. A key assumption is that queries can be reliably decomposed into concepts, polarities, and operators in a way that matches the later truth-functional execution. The paper gives meaningful support for this, but it remains one of the most important assumptions in the overall logic of the method, even though it is mentioned in the Limitation section.
3. The paper moves fairly quickly from "atomic evidence is often reliable" to "the failure is mainly in execution." That step is reasonable, but the main text could separate execution failures from residual evidence-extraction failures more cleanly.
4. The evidence from Table 1, Figures 2-5, and the appendix diagnostics supports the authors' claim, but it still reads more like support for an interface-dominant explanation than a full exclusion of alternative contributors. The conclusion is defensible, but the wording could be more measured.
5. I still have to say, all figures in the manuscript should be polished, like font style and size.

---

> ### Author Rebuttal · Authors · 2026-03-30
>
> We thank the reviewer for the careful reading. We address their questions and comments below.
>
> **Scope of the bottleneck claim (W1, W3, W4).** The reviewer's distinction between an **interface-dominant** and an **interface-exclusive** explanation captures the intended scope of our claim. We agree that the paper's evidence supports the former, not the latter. The paper does not prove that execution is the only coupled factor. Rather, it provides strong empirical evidence that it is a major one in the studied Boolean setting. The span-residual decomposition, operator-gain analysis, and amplification experiment each contribute to this case. We will revise the abstract, introduction, and conclusion to state this as a scoped empirical finding rather than a universal claim.
>
> To strengthen this interpretation, we grouped FACTOR-Bench samples by the minimum CLIP concept-detection AUC among involved concepts (the "weakest link," since LCSE uses all concept scores in the correction):
>
> | Concept AUC bin | n | Holistic (%) | LCSE (%) | Δ (pp) |
> |-----------------|---|-------------|---------|--------|
> | High (>0.90) | 417 | 57.8 | 90.4 | +32.6 |
> | Medium (0.75-0.90) | 479 | 61.4 | 85.2 | +23.8 |
> | Low (<0.75) | 204 | 52.0 | 76.0 | +24.0 |
>
> When atomic evidence is strong, LCSE reaches **90.4%**, which suggests that the remaining errors in this regime are better explained by evidence quality than by operator execution. When evidence is weaker, performance drops predictably to 76.0%, but still exceeds holistic by +24pp. This pattern gives us stronger empirical confidence that LCSE is resolving a substantial execution bottleneck and that remaining errors increasingly track evidence quality rather than operator execution.
>
> **Decomposition assumption (W2).** The reviewer is right that this assumption is central to the method and deserves more prominence than its current placement in the Limitations section. Parsing is a modular upstream component rather than the main methodological contribution, but its quality matters for end-to-end use. We therefore evaluate under both conditions: oracle parses on FACTOR-Bench to isolate execution from parser noise, and real GPT-4.1-mini parses on NegBench and retention. When we replace oracle with LLM parses on FACTOR-Bench, accuracy drops by only 0.3pp with 0 wrong-sign corrections across the 1,100 operator samples in FACTOR-Bench. We will make both the assumption and the empirical evidence supporting it more prominent in the main text.
>
> **Figure polish (W5).** We agree and will standardize font sizes, label styles, and axis formatting across all figures in the revision.

---

> > ### Author Rebuttal · Reviewer_dm4j · 2026-04-03
> >
> > I don't have further questions.

---

> > > ### Author Response · Authors · 2026-04-06
> > >
> > > Thank you for your time and your constructive feedback. We are glad that your concerns are fully resolved.

---

### Decision · Program_Chairs · 2026-04-30

**Decision:**

Accept (regular)

**Comment:**

This paper studies the limitations of dual-encoder vision-language models (VLMs): if a query has logical operators, the cosine similarity aggregates concept evidence in a "bag-of-concepts" manner and does not properly enforce logical constraints. To tackle the issue, this paper proposes a factorized inference, Logic-Constrained Score Editing (LCSE), which separates evidence extraction from constraint execution. This paper also introduces FACTOR-Bench, an evaluation benchmark for logical compositionality in retrieval.

The reviewers found that the problem is well-motivated and well-supported, the analyses are novel and rigorous, and the empirical gain looks significant. On the other hand, the reviewers raised some concerns, such as the limited scalability (due to the computation cost), the sensitivity to the hyperparameter, the reliance on an LLM parser, and the limited complexity of compositional logics used in this paper.

After the discussion period, all reviewers acknowledged that their concerns were fully addressed and reached a positive consensus. I also agree with the reviewers' opinions.

Overall, this paper introduces an interesting and important problem, which is well-supported by additional analyses. The empirical gain also looks promising. I think this paper exceeds the acceptance bar of ICML; hence, I recommend accepting this paper.